# Autophagy within the mushroom body protects from synapse aging in a non-cell autonomous manner

Anuradha Bhukel[1,2], Christine Brigitte Beuschel [1,2], Marta Maglione[1,2], Martin Lehmann[3], Gabor Juhász[4], Frank Madeo[5,6] & Stephan J. Sigrist[1,2]

Macroautophagy is an evolutionarily conserved cellular maintenance program, meant to protect the brain from premature aging and neurodegeneration. How neuronal autophagy, usually loosing efficacy with age, intersects with neuronal processes mediating brain maintenance remains to be explored. Here, we show that impairing autophagy in the *Drosophila* learning center (mushroom body, MB) but not in other brain regions triggered changes normally restricted to aged brains: impaired associative olfactory memory as well as a brain-wide ultrastructural increase of presynaptic active zones (metaplasticity), a state non-compatible with memory formation. Mechanistically, decreasing autophagy within the MBs reduced expression of an NPY-family neuropeptide, and interfering with autocrine NPY signaling of the MBs provoked similar brain-wide metaplastic changes. Our results in an exemplary fashion show that autophagy-regulated signaling emanating from a higher brain integration center can execute high-level control over other brain regions to steer life-strategy decisions such as whether or not to form memories.

---

[1] Institute for Biology/Genetics, Freie Universität Berlin, Takustr. 6, 14195, Berlin, Germany. [2] NeuroCure, Charité, Charitéplatz 1, 11007 Berlin, Germany. [3] Leibniz Forschungsinstitut Für Molecular Pharmakologie, Campus Berlin-Buch, Robert-Roessle-Str. 10, 13125 Berlin, Germany. [4] Department of Anatomy, Cell and Developmental Biology, Eötvös Loránd University, Pázmány, s. 1/C. 6.520, Budapest H-1117, Hungary. [5] Institute for Molecular Biosciences, NAWI Graz, University of Graz, Humboldtstrasse 50/EG, 8010 Graz, Austria. [6] BioTechMed Graz, 8010 Graz, Austria. These authors contributed equally: Anuradha Bhukel, Christine Brigitte Beuschel. Correspondence and requests for materials should be addressed to S.J.S. (email: stephan.sigrist@fu-berlin.de)

Macroautophagy (henceforth, autophagy) is a process of cellular self-digestion in which portions of the cytoplasm and even whole organelles are sequestered in double-membrane or multi-membrane vesicles (autophagosomes), and then delivered to lysosomes for bulk degradation. Autophagy lately came in the focus for its apparently crucial role in the aging and neurodegeneration process[1–3]. Compromised efficacy of autophagy is suspected to contribute to brain aging, and in reverse rejuvenating autophagy in aging neurons is considered a promising strategy to restore cognitive performance[4–6]. Autophagosome biogenesis takes place mainly in distal axons, close to presynaptic specializations[7,8] and is suggested to reduce with age[9]. Retrograde transport of autophagosomes might play a role in neuronal signaling processes, promoting neuronal complexity and preventing neurodegeneration[4–6].

Changes in synaptic strength with increased or decreased synaptic activity (synaptic plasticity) are considered to be the core process regulating memory formation across all model systems investigated[10,11]. Deficits in plasticity might thus be of particular importance for age-induced cognitive decline. In fact, instead of emphasizing the loss of neurons, studies in several models now point towards rather subtle age-related synaptic alterations in the hippocampus and other parts of the cortical brain as being associated with age-associated cognitive decline[12–14]. However, causal connections between synaptic changes and age-associated cognitive decline remain to be established. In short, the challenge is to identify the mechanisms of protein homeostasis that are active in neurons, and to understand how these mechanisms intersect with the multiple aspects of neuron function and plasticity over a lifetime.

Here we show that genetically impairing autophagy within the major learning-related brain-center of *Drosophila*, called mushroom body (MB), sufficed to trigger brain-wide changes in presynaptic organization in a non-cell autonomous manner. The occurrence of this brain-wide presynaptic metaplasticity was invariably connected to the absence of the specific component of aversive olfactory memory, which normally only declines in the course of the aging process. In contrast, attenuating autophagy in other brain centers was without any measureable effect on synaptic metaplasticity, not even locally in neurons under direct genetic manipulation. We further found signaling of the metabolism-related NPY-type neuropeptide within the MBs (but not MB synaptic vesicle release or excitability) to be important to protect the brain from premature metaplasticity and consequently a decay of memory formation capability. From a broader perspective, our results provide evidence for a high-level control by a brain integration center (here the MB), whose autophagy status can seemingly tune the overall information processing strategy of an entire brain.

## Results

**Neuronal attenuation of autophagy impairs age-sensitive memory.** In order to attenuate autophagy specifically in the neurons of the *Drosophila* brain, we expressed RNA interference (RNAi) constructs targeting core components of autophagy machinery via a pan-neuronal driver line (*elav*-Gal4)[15,16]. One diagnostic feature of autophagic efficacy is the degradation of the ubiquitin-binding scaffold protein p62/SQSTM1. Ubiquitinated proteins meant for autophagic degradation are positive for p62/SQSTM1[17] (*Drosophila* homolog: Ref(2)p), which due to its interaction with LC3/ATG8 is degraded via autophagy[18]. Therefore, lack of autophagy leads to an accumulation of p62 aggregates, while induction of autophagy reduces p62. We hence used p62 as read-out to in the *Drosophila* brain screen across RNAi lines directed against the expression of autophagy core

components (Table 1). After depletion of various such ATG proteins (essential for autophagy induction) we tested the fly brains for p62 accumulation. Throughout this study, animals of 10 days of adult life were tested, with this age chosen as a compromise between there being enough time for effects (such as p62 accumulation) to build up, but the effects of our manipulations not already being overshadowed by upcoming effects of aging.

Among the lines tested, RNA lines targeting *atg9* and *atg5*, respectively, yielded robust elevation of p62/Ref(2)p aggregates as evident in immunostainings of adult *Drosophila* brains (Fig. 1a–f). This increased accumulation of p62 was also clearly visible in Western blots of brain homogenates (Fig. 1g). We further analyzed animals with pan-neuronal RNA against Atg5 and Atg9. *atg5* transcript levels in morphologically isolated brains of the *elav/atg5*-RNAi were reduced by almost 40% compared to controls (relative fold change: $0.5964 \pm 0.1034$. $**p < 0.005$, Paired *t*-test, $n = 7$). *atg9* transcript levels in morphologically isolated brains of the *elav/atg9*-RNAi were reduced by almost 50% compared to controls (relative fold change: $0.5319 \pm 0.1261$. $**p < 0.005$, Paired *t*-test, $n = 7$). The above-mentioned KD efficiencies are likely an underestimation of transcript KD as not all brain cells, particularly glia, are targeted by this pan-neuronal driver line. Atg5 forms a complex with Atg12 and Atg16, which acts as an E3 ligase in the lipidation of Atg8 (LC3) to promote the elongation of the autophagosomal membranes[19]. A deficiency of Atg5 should inhibit the lipidation process. The *elav/atg5*-RNAi brains as expected showed an absence of second band in Western blot corresponding to lipidated Atg8 (Fig. 1g). Atg9, on the other hand, is a trans-membrane protein that delivers membrane lipids to the growing autophagosomes and is not essential for lipidation of Atg8[20]. In fact, the lipidation of Atg8 was not attenuated upon *atg9* knockdown (KD) (Fig. 1g). Thus, we went on to use *atg9* and *atg5* KD in neurons as two per se independent molecular scenarios, which, however, converge onto the impairment of (macro)autophagy. We compared these two situations in order to identify the generic roles of autophagy for neuronal and cognitive maintenance of the fly brain.

Misregulation of neuronal autophagy might per se also trigger apoptosis[21], a process associated with age in *Drosophila*[22]. To investigate whether the manipulation of autophagy using these two RNAi constructs would cause any ectopic apoptosis, we immunostained 10-day-old fly brains for Annexin V and activated Death caspase-1 (Dcp-1), respectively, to detect apoptotic cells. Annexin V binds to phosphatidylserine, a marker of apoptosis when it is on the outer leaflet of the plasma membrane[23]. Dcp-1 is a commonly used marker for cells undergoing apoptosis in *Drosophila*[24]. While we could visualize the previously described age-induced increase of apoptotic cells, no such increase was observed in *elav/atg5*-RNAi brains or *elav/atg9*-RNAi brains (Supplementary Fig. 1). Neither in electron microscopic analysis did we observe apoptosis-typical signs of nuclear condensation, boundary aggregation and splitting, and DNA fragmentation in these animals. Moreover, if apoptosis was misregulated in the CNS upon inhibition of autophagy, an altered count of cell bodies should be expected. We thus, further quantified the number of cell bodies of MB intrinsic neurons and found no difference between control and upon inhibition of autophagy in MB (Supplementary Fig. 2).

We went on testing whether a pan-neuronal reduction of autophagy would affect olfactory memory formation. Innate smell scores were unaffected in both *elav/atg5*-RNAi and *elav/atg9*-RNAi (Table 2). Short-term aversive olfactory memory scores were, however, significantly reduced after knocking down either *atg*-gene with *elav*-gal4 (Fig. 2a, b). Mid-term memory (MTM), measured one hour after conditioning, has been reported to decline with age[25–27]. Notably, pan-neuronal inhibition of either Atg9 or Atg5 led to an early reduction in MTM scores in

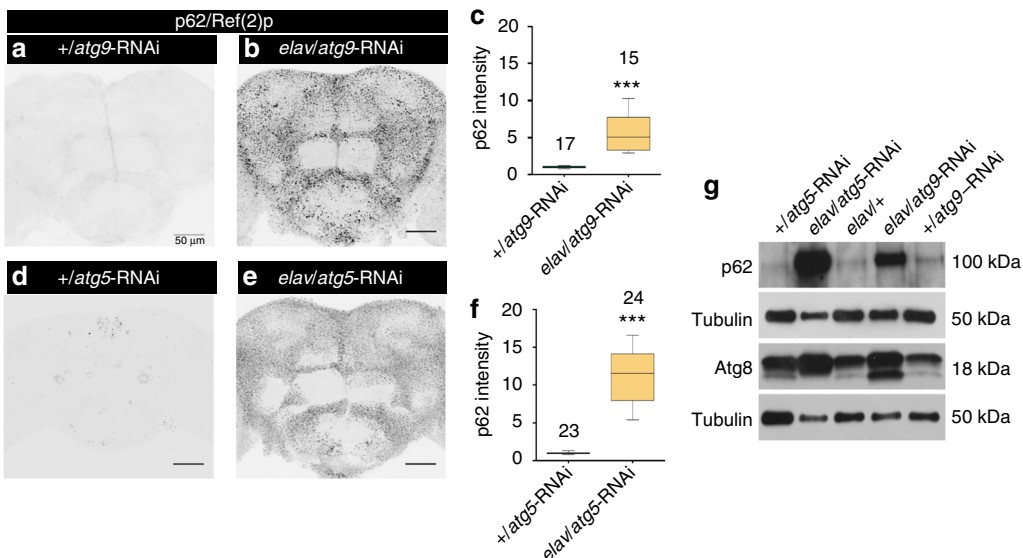

**Fig. 1** Neuronal knockdown of atg9 or atg5 suppresses autophagy in the brain. Adult brain, 10 days (**a**) +/atg9-RNAi, (**b**) elav/atg9-RNAi, (**d**) +/atg5-RNAi, and (**e**) elav/atg5-RNAi immunostained for p62/Ref(2)p. Scale bar: 50 μm. **c** Quantification of p62/Ref(2)p intensity within the central brain region normalized to control (n = 15–17 independent brains; ***p < 0.001; Mann–Whitney U-test). **f** Quantification of p62/Ref(2)p intensity within the central brain region normalized to control (n = 23–24 independent brains; ***p < 0.001; Mann–Whitney U-test). **g** Selective autophagy marker p62/Ref(2)p and autophagosome-associated protein Atg8a accumulate in protein homogenate prepared from the brain of elav/atg9-RNA and elav/atg5-RNAi compared to respective controls. An amount equivalent to 1 brain was loaded onto the gel. Lipidated Atg8a (Atg8a-II) is missing upon KD of atg5 but not when atg9 is knocked down. In the box and whisker plots, the middle line of a box represents the median (50th percentile) and the terminal lines of a box represent the 25th and 75th percentile. The whiskers represent the lowest and the highest value

10-day-old animals already (Fig. 2c, d). This decline in MTM scores was indeed quantitatively similar to what we observed in aged (30 days) control animals (Supplementary Fig. 3).

**Autophagy in the MB controls age-sensitive memory**. We next addressed whether specific neuron types within the *Drosophila* brain would be of particular importance in mediating the role of autophagy in protecting aversive olfactory memories. Olfactory sensory neurons (OSNs) transmit their information to projection neurons (PNs) in the central brain, with their performance being critical for smell response and learning of olfactory cues[28–30]. The majority of PNs can be labeled with the transgenic reporter line *gh146*-Gal4[28]. Both *gh146/atg9*-RNAi and *gh146/atg5*-RNAi had p62 aggregates within PN cell bodies (Supplementary Fig. 4). Innate smell scores but also aversive memory scores remained unaffected in these flies (Supplementary Fig. 4, Table 2).

We then targeted the MB, a higher integration center critical for memory acquisition and storage, using two independent driver lines (*ok107*-Gal4; *vt30559*-Gal4)[31,32]. Combination of either driver line with either the *atg5* or the *atg9*-RNAi inducible transgene provoked a strong buildup of both p62 and Atg8a in the cell bodies of the MB intrinsic neurons (Kenyon cells; KC) but did not affect levels of downstream effector Syntaxin-17 (Supplementary Figs. 5–7). The innate smell scores remained comparable between control and MB-specific KD of either *atg*-gene (Table 2). Importantly, however, KD of the *atg5/atg9* core autophagy components using these MB expressing drivers significantly impaired STM scores (Fig. 2e–g).

We tested MTM for *ok107/atg9*-RNAi. In fact, MTM scores were significantly reduced here as well (Fig. 2h). MTM, via genetic or pharmacological interference[33–35], can be further dissected into two components: anesthesia-sensitive memory (ASM) and anesthesia-resistant memory (ARM). ASM is calculated by subtracting ARM scores, measured after amnestic cooling, from MTM. ASM is considered to be the precursor of

long-term consolidated memory (LTM)[36]. Most important for this context, it is the ASM, unlike the ARM, which has been shown to be strongly impaired with aging[25,27,37]. We compared ARM and ASM scores for pan-neuronal (*elav*-Gal4) and MB-specific (*ok107*-Gal4) KD of *atg9*. Both, pan-neuronal and equally MB-specific KD significantly reduced the ASM but not the ARM component of the MTM (Fig. 2i–l). Moreover, this decline in ASM scores upon pan-neuronal inhibition of autophagy was indeed quantitatively similar to what we observed in aged (30 days) control animals (Supplementary Fig. 3). Thus, KD of a core autophagy component within the MBs suffices to mimic the usual age-induced specific decay of the ASM already in young animals. In other words, keeping proper autophagic function within the MB seems of importance to prevent premature decline of the age-sensitive MTM precursor. The fact that innate smell scores and principal locomotion behavior of these flies remained unaffected speaks in favor of a specific deficit and not a generic degeneration after our comparatively mild KD.

**Impairing MB autophagy increases Bruchpilot brain-wide**. Presynaptic plasticity supports memory formation in *Drosophila*[38]. Previously, we analyzed age-induced changes in the ultrastructural, molecular, and functional organization of presynapses within the olfactory system of flies. We here found that aging is associated with a brain-wide increase in the synaptic staining of active zone (AZ) master scaffold protein Bruchpilot (BRP)[39]. Young *atg7* generic null mutant flies mimicked aged flies in showing both this increase in BRP staining, as well as an inability to form olfactory memories[37,39]. Pan-neuronal KD of either *atg5* or *atg9* triggered a similar increase in BRP[Nc82] label brain-wide (Fig. 3a–c, g–I, Supplementary Fig. 8). Thus, the autophagic status of neurons (and apparently not other cell types such as glia) seems to control the presynaptic BRP level and consequently AZ scaffold structure.

As before with analyzing olfactory memory, we again restricted our manipulations to defined neuron populations of the olfactory

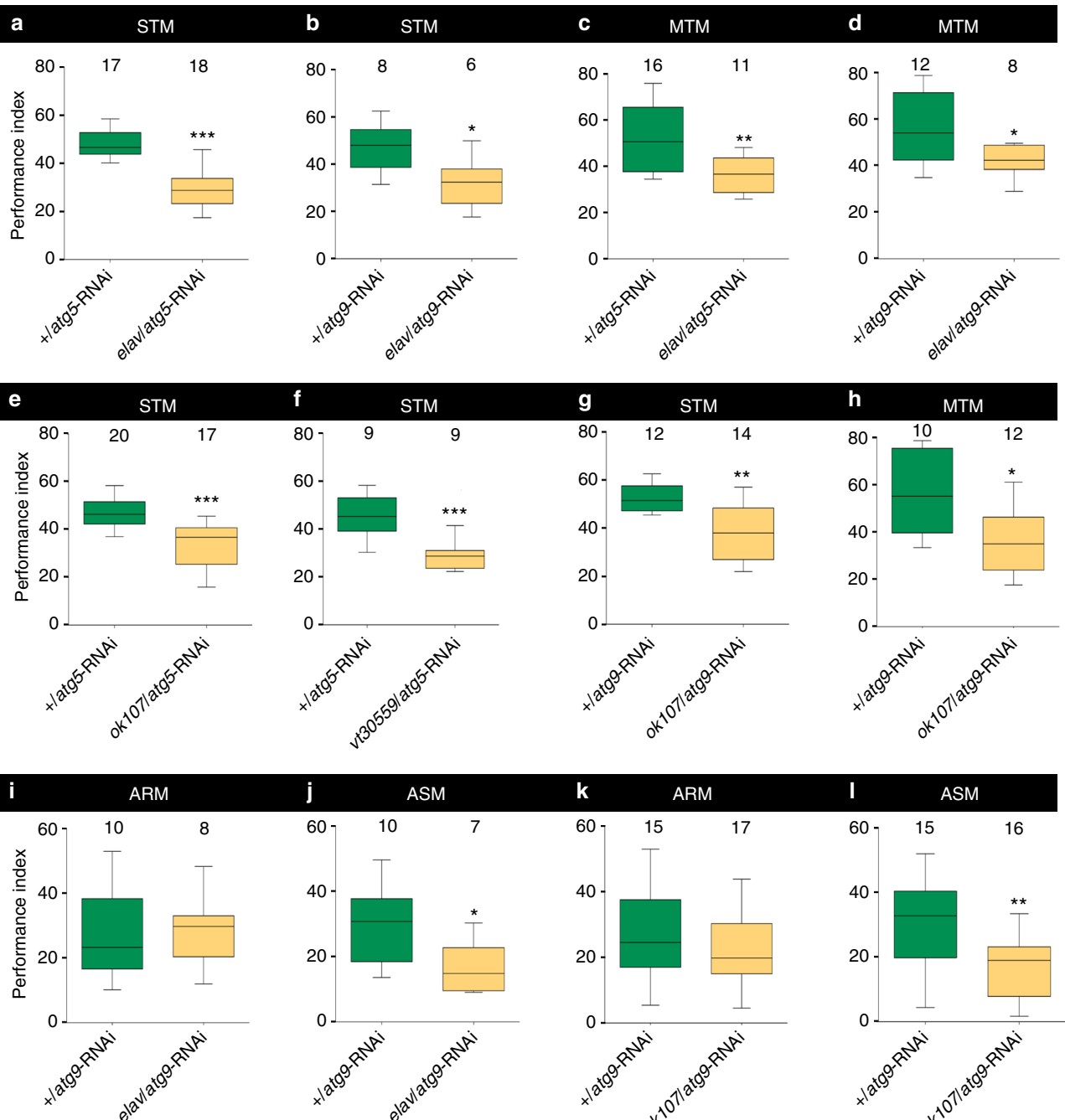

**Fig. 2** Neuronal inhibition of autophagy impairs age-sensitive olfactory memory. **a** Aversive associative memory performance 3 min after training (short-term memory, STM) markedly reduced in *elav/atg5*-RNAi compared to +/*atg5*-RNAi ($n = 17$–18; ***$p < 0.001$; Mann–Whitney $U$-test). **b** STM significantly reduced in *elav/atg9*-RNAi compared to +/*atg9*-RNAi ($n = 6$–8; *$p < 0.05$; Mann–Whitney $U$-test). **c** Aversive associative memory performance at 1 h after training (mid-term memory; MTM) of *elav/atg5*-RNAi declined significantly compared to +/*atg5*-RNAi ($n = 11$–16; **$p < 0.01$; Mann–Whitney $U$-test). **d** MTM significantly reduced in *elav/atg9*-RNAi flies compared to +/*atg9*-RNAi ($n = 8$–12; *$p < 0.05$; Mann–Whitney $U$-test). **e** STM significantly reduced in *ok107/atg5*-RNAi flies compared to +/*atg5*-RNAi ($n = 17$–20; ***$p < 0.001$; Mann–Whitney $U$-test). **f** STM significantly reduced in *vt30559/atg5*-RNAi flies compared to +/*atg5*-RNAi ($n = 9$; ***$p < 0.0001$; Mann–Whitney $U$-test). **g** STM significantly reduced in *ok107/atg9*-RNAi flies compared to +/*atg9*-RNAi ($n = 12$–14; **$p < 0.01$; Mann–Whitney $U$-test). **h** MTM significantly reduced in *ok107/atg9*-RNAi flies compared to +/*atg9*-RNAi ($n = 10$–12; *$p < 0.05$; Mann–Whitney $U$-test). **i** Aversive associative memory performance at 1 h after training, anesthesia-resistant memory (ARM) and **j** anesthesia-sensitive memory (ASM) of *elav/atg9*-RNAi compared to +/*atg9*-RNAi ($n = 7$–10; *$p < 0.05$; ns$p > 0.5$; Mann–Whitney $U$-test). **k** ARM and **l** ASM of *ok107/atg9*-RNAi compared to +/*atg9*-RNAi ($n = 15$–17; **$p < 0.01$; ns$p > 0.5$; Mann–Whitney $U$-test). In the box and whisker plots, the middle line of a box represents the median (50th percentile) and the terminal lines of a box represent the 25th and 75th percentile. The whiskers represent the lowest and the highest value

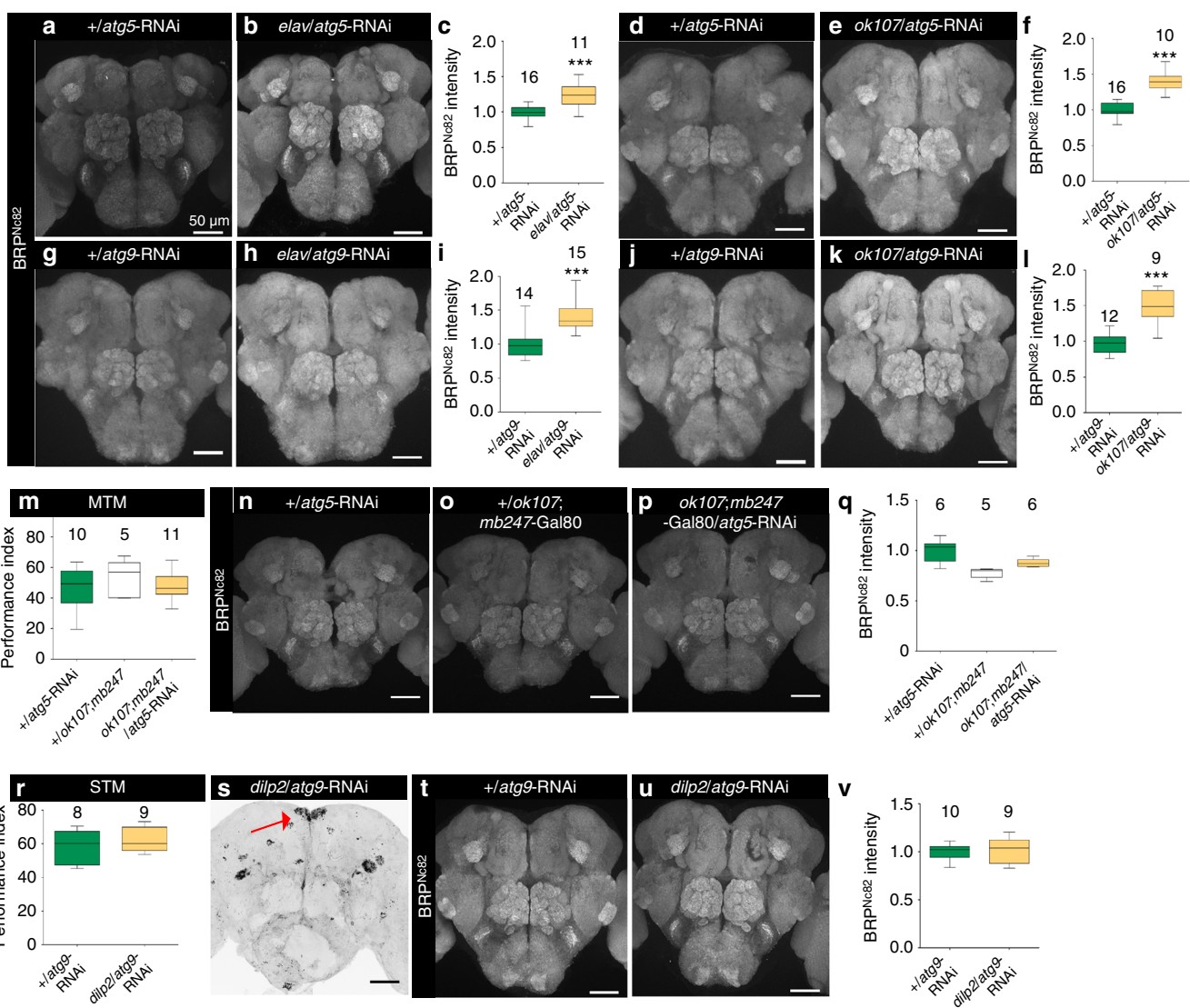

**Fig. 3** Mushroom body-specific suppression of autophagy induces brain-wide increase of Bruchpilot. Adult brain, 10 days (**a**) +/*atg5*-RNAi and (**b**) *elav*/*atg5*-RNAi immunostained for BRP$^{Nc82}$. Scale bar: 50 μm. **c** Quantification of BRP$^{Nc82}$ intensity within the central brain region normalized to control (*n* = 11–16 independent brains; \*\*\**p* < 0.001; Mann–Whitney *U*-test). Adult brain, 10 days (**d**) +/*atg5*-RNAi and (**e**) *ok107*/*atg5*-RNAi immunostained for BRP$^{Nc82}$. Scale bar: 50 μm. **f** Quantification of BRP$^{Nc82}$ intensity within the central brain region normalized to control (*n* = 10–16 independent brains; \*\*\**p* < 0.001; Mann–Whitney *U*-test). Adult brain, 10 days (**g**) +/*atg9*-RNAi and (**h**) *elav*/*atg9*-RNAi immunostained for BRP$^{Nc82}$. Scale bar: 50 μm. **i** Quantification of BRP$^{Nc82}$ intensity within the central brain region normalized to control (*n* = 14–15 independent brains; \*\*\**p* < 0.001; Mann–Whitney *U*-test). Adult brain, 10 days (**j**) +/*atg9*-RNAi and (**k**) *ok107*/*atg9*-RNAi immunostained for BRP$^{Nc82}$. Scale bar: 50 μm. **l** Quantification of BRP$^{Nc82}$ intensity within the central brain region normalized to control (*n* = 9–12 independent brains; \*\*\**p* < 0.001; Mann–Whitney *U*-test). **m** MTM of +/*atg5*-RNAi, +/*ok107*;*mb247*-Gal80 and *ok107*;*mb247*-Gal80/*atg5*-RNAi (*n* = 10–15; $^{ns}p$ > 0.5; One-way ANOVA; Kruskal–Walis post-test). Adult brain, 10 days (**n**) +/*atg5*-RNAi, (**o**) +/*ok107*;*mb247*-Gal80, and (**p**) *ok107*; *mb247*-Gal80/*atg5*-RNAi immunostained for BRP$^{Nc82}$. Scale bar: 50 μm. **q** Quantification of BRP$^{Nc82}$ intensity within the central brain region normalized to +/*atg5*-RNAi (*n* = 5–6 independent brains; $^{ns}p$ > 0.5; One-way ANOVA; Kruskal-Walis test). **r** STM of +/*atg9*-RNAi and *dilp2*/*atg9*-RNAi (*n* = 8–9; $^{ns}p$ > 0.5; Mann–Whitney *U*-test). Adult brain, 10 days (**s**) *atg9*-RNAi/+ immunostained for p62/Ref(2)p. Scale bar: 50 μm. Adult brain, 10 days (**t**) +/*atg9*-RNAi and (**u**) *dilp2*/*atg9*-RNAi immunostained for BRP$^{Nc82}$. Scale bar: 50 μm. **v** Quantification of BRP$^{Nc82}$ intensity within the central brain region normalized to control (*n* = 11–16 independent brains; $^{ns}p$ > 0.5; Mann–Whitney *U*-test). $^{ns}p$ > 0.01, \*\*\**p* < 0.0001. In the box and whisker plots, the middle line of a box represents the median (50th percentile) and the terminal lines of a box represent the 25th and 75th percentile. The whiskers represent the lowest and the highest value

system. Surprisingly, when suppressing either *atg5* or *atg9* with either driver line expressing in the MB (*ok107*-Gal4 or *vt30559*-Gal4), changes in BRP intensity clearly extended over the expression domain of these drivers. In fact, BRP in all cases appeared to be elevated over the entire central brain (Fig. 3d–f, j–l, Supplementary Fig. 9). In contrast, autophagy suppression in the PNs did not influence the BRP staining to any measureable extent (Supplementary Fig. 4).

To directly test whether MB-specific expression plays an essential role in the non-cell autonomous spreading of this synaptic phenotype in the *Drosophila* brain, we prevented *atg-5* KD (via *ok107*-Gal4) in the MBs by simultaneously expressing a Gal4 repressor, Gal80, under control of an independent MB-specific promoter, *mb247*. We found that MTM was restored when Gal4 activity was simultaneously blocked in MBs with *mb247*-Gal80 (Fig 3m). Concurrently, the brain-wide increase in

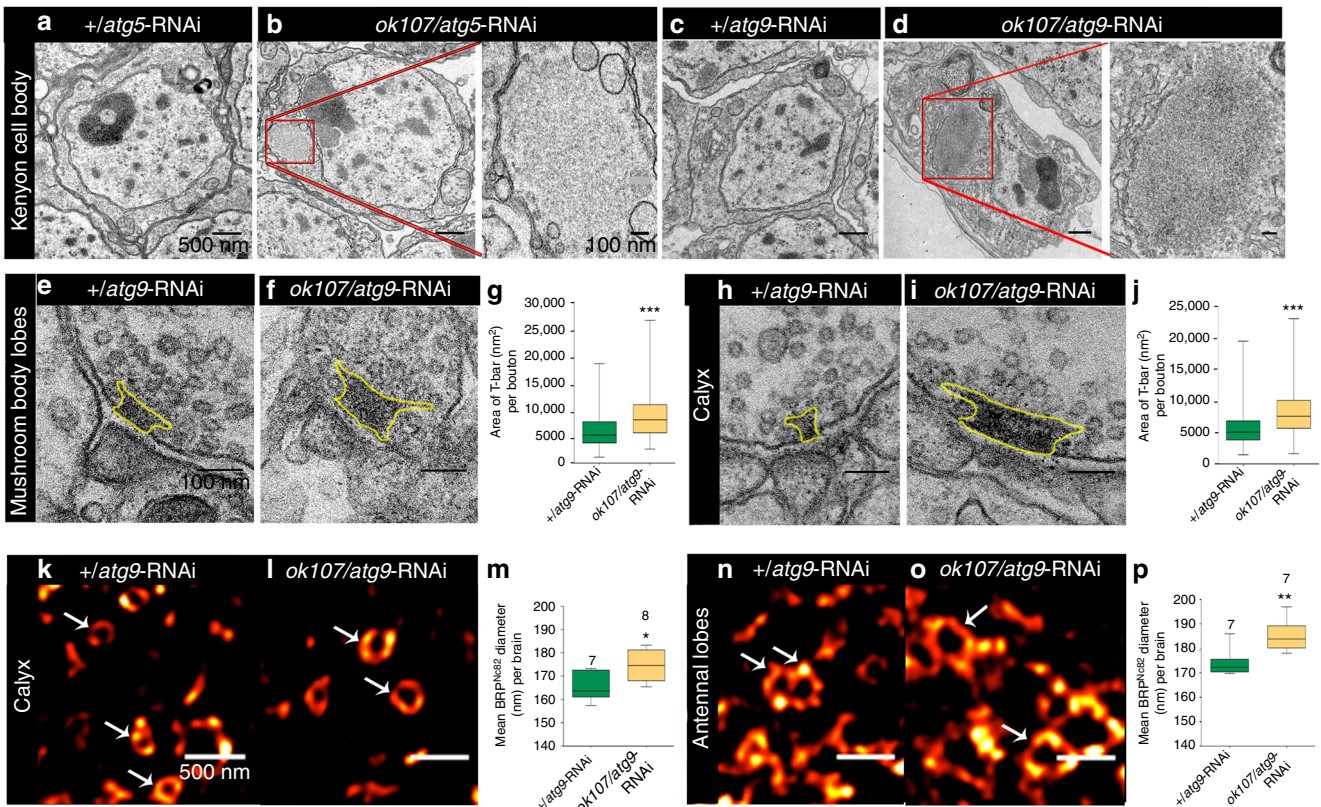

**Fig. 4** Autophagy suppression in Mushroom body causes age-typical ultrastructural presynaptic phenotypes. Electron micrograph of Kenyon cell bodies in (**a**) +/*atg5*-RNAi and (**b**) *ok107*/*atg5*-RNAi animals. Scale bar: 500 nm. Aggregates build up in *ok107*/*atg5*-RNAi animals (highlighted in red box). Scale bar: 100 nm. Electron micrograph of Kenyon cell bodies in (**c**) +/*atg9*-RNAi and (**d**) *ok107*/*atg9*-RNAi animals. Scale bar: 500 nm. Aggregates build up in *ok107*/*atg9*-RNAi animals (highlighted in red box). Scale bar: 100 nm. Electron micrograph showing presynaptic specializations at KC-to-MBON synapses in medial MB lobes of (**e**) +/*atg9*-RNAi and (**f**) *ok107*/*atg9*-RNAi animals (outlined as yellow). Scale bar: 100 nm. **g** Quantification of average T-bar size in +/*atg9*-RNAi and *ok107*/*atg9*-RNAi animals ($n = 240$ electron micrographs across 3–5 independent animals, with at least 20 T-bars per animals; ***$p < 0.001$; Mann–Whitney $U$-test). Electron micrograph showing presynaptic specializations at PN-to-KC synapses in Calyx of (**h**) +/*atg9*-RNAi and (**i**) *ok107*/*atg9*-RNAi animals (outlines as yellow). Scale bar: 100 nm. **j** Quantification of average T-bar size in +/*atg9*-RNAi and *ok107*/*atg9*-RNAi animals ($n = 268$ electron micrographs across four independent animals, with atleast 20 t-bars per animal; ***$p < 0.001$; Mann–Whitney $U$-test). STED images of BRP spots reveal ring-shaped structures (pointed by arrowheads) within the (**k**, **l**) Calyx and (**n**, **o**) antennal lobe of +/*atg9*-RNAi and *ok107*/*atg9*-RNAi. Scale bar: 500 nm. **m** Comparison of BRP ring diameter between Calyx of +/*atg9*-RNAi and Calyx of *ok107*/*atg9*-RNAi (total of 2931 BRP rings across 16 independent animals, with at least five rings per animals; *$p < 0.05$; Mann–Whitney $U$-test). **p** Comparison of BRP ring diameter between antennal lobe of +/*atg9*-RNAi and antennal lobe of *ok107*/*atg9*-RNAi (total of 1327 BRP rings across 15 independent animals, with at least five rings per animal; **$p < 0.01$; Mann–Whitney $U$-test). In the box and whisker plots, the middle line of a box represents the median (50th percentile) and the terminal lines of a box represent the 25th and 75th percentile. The whiskers represent the lowest and the highest value

BRP signal (previously observed in *ok107*/*atg5*-RNAi flies) was no longer detected when Gal4 was simultaneously blocked in MBs with Gal80 (Fig. 4n–q). We hence conclude that autophagy in the MBs plays a crucial role in determining the non-cell autonomous spreading of memory-aversive synaptic phenotypes in the *Drosophila* brain. A brain wide increase of synaptic BRP levels thereby is associated with decreasing ASM type memory.

Besides from expressing in MBs, the enhancer trap line *ok107*-Gal4 has been reported to express in median neurosecretory cells (mNSCs), the designated insulin-producing cells (IPCs). The assumed dendrites of IPCs are located in the *Pars Intercerebralis*, dorsally in the protocerebrum[40–42]. Altered insulin-type signaling has in fact been linked with non-cell autonomous effects of autophagy[43,44]. We, therefore, performed *atg9* KD in mNSCs using *dilp2*-Gal4[45]. As expected, *atg9* KD in these cells resulted in a strong buildup of p62 aggregates in the targeted neuron populations (Fig. 3s, arrows). However, neither memory scores (Fig. 3r) nor BRP staining levels (over the central brain) (Fig. 3t–v) were significantly changed compared to controls.

Thus, autophagy suppression specifically in *Pars Intercerebralis* is obviously not sufficient for the observed non-cell autonomous spreading of synaptic phenotypes and a subsequent defect in memory formation. Together, we conclude that the autophagic status of the MB broadly controls presynaptic metaplasticity in an obviously non-cell autonomous manner.

**MB autophagy causes brain-wide presynaptic phenotypes.** Complete genetic abrogation of (macro)autophagy after KD of autophagy core components in mice provokes massive neuron loss in cerebral and cerebellar regions in the course of months[46,47]. We however, did not observe any signs of neuronal death in fly CNS after MB-specific attenuation (but likely not complete abrogation) of autophagy. While KD of either *atg5* (Fig. 4a, b) or *atg9* (Fig. 4c, d) lead to a build-up of aggregates in MB Kenyon cell bodies, we did not observe any signs of neuro-degeneration in electron micrographs acquired for these neurons. Moreover, the MBs seemingly developed normally in these animals (Supplementary Fig. 10).

We next addressed the presynaptic status of AZs within and outside the MB after impairing autophagy within the MBs. To this end, we subjected the brains of *ok107/atg9* KD to transmission emission electron microscopy (EM) and super-resolution light microscopy (STED). The pre-synaptic AZ scaffold exhibits an electron dense structure in EM[48]. At *Drosophila* synapses, the AZ scaffold appears as a T-shaped structure, hence named T-bar. While principle synapse organization appeared normal in the *ok107/atg9*-RNAi, T-bars within the MB intrinsic KCs appeared more prominent. Quantification found them to cover a significantly larger area in transmission EM sections (Fig. 4e–g). As said earlier, we observed an increase of synaptic BRP label beyond the MB region. Thus, we also analyzed synapses outside the MB in ultrastructural detail. PN boutons form AZs in the Calyx region on KC. Again, principle organization of boutons seemed not to differ between the MB-specific *atg9* KD and the control. However, T bar areas were found similarly increased here as well as at KC output synapses (Fig. 4h–j). Using the T-bars to identify AZs, we noticed a slight increase in AZ density normalized to synaptic bouton area in both Calyx and MB after MB-specific attenuation of *atg9* (Supplementary Fig. 11). Given the increased average size of T bar scaffolds in this genotype, this result might also reflect increased detection sensitivity for AZs.

Stimulated emission depletion microscopy (STED) previously allowed us to unmask the nano-architecture of *Drosophila* AZs. Here, $Ca^{2+}$ channels are concentrated in the AZ center and BRP organizes a scaffold around these $Ca^{2+}$ channels to provide the slots for the SV release[49–51]. As per AZ BRP signals scale with per AZ release of synaptic vesicles[52,53], STED-derived measurements of AZs allow to estimate the functional status of individual AZs. As STED currently is limited to rather superficial neuropil layers, we subjected calyx synapses to STED. In fact, BRP scaffold diameters were increased at PN output synapses of the MB-specific *atg9* KDs (Fig. 4k–m). Finally, we also chose olfactory synapses of OSNs in the antennal lobe, which are two relay synapses apart from the MB (and also superficially located). Olfactory receptor neurons here form multiple connected AZ T-bars, which in planar STED images appear as concatenated rings[51]. Also at these synapses, BRP scaffold diameters were equally increased after MB-specific KD of *atg9* (Fig. 4n–p). Thus, we have to conclude that the autophagic status of the MB somehow can influence the ultrastructural and consequently likely the functional status of synapses throughout the olfactory system and beyond. In other words, MB-specific attenuation of autophagy can trigger presynaptic metaplasticity in a non-cell autonomous manner.

As far as we could see, the synaptic phenotype observed throughout the *Drosophila* brain after MB-specific attenuation of autophagy appeared identical to the brain-wide ultrastructural phenotype of aged flies which we reported previously[39]. Thus, our ultrastructural analysis indicates that autophagic decline within the MB could be of major importance for driving the age-induced decline of learning and memory processes.

In additional experiments, we sought to limit the *atg5/9* KD to adult life only by combining the RNAi line/driver combinations with a temperature-sensitive Gal80 construct, and shifting these flies to non-permissive temperature (29 °C) after eclosion. Unfortunately, however, this approach did not result in a measurable accumulation of p62 in KCs, likely reflecting a slow turnover of already developmentally expressed Atg5 or Atg9. While we currently cannot differentiate between contributions of autophagy suppression in pre-hatching and post-hatching phase, our arguments concerning a non-cell autonomous role of the MB remain unaffected. We finally addressed putative mechanisms through which the MB might steer presynaptic metaplasticity in an essentially brain-wide fashion.

**Neuropeptide F (NPF) from MB protects from metaplasticity.**
We first asked whether modulations of fast synaptic neurotransmission of MB neurons might per se be able to provoke a brain-wide spreading of synaptic phenotypes similarly as observed after MB-specific attenuation of autophagy. To this end, we silenced the synaptic release from the MB neurons throughout development until adult day 10 by expressing a temperature-sensitive Dynamin allele (*shibire*[ts])[54] within the MBs of animals raised at restrictive temperature (29 °C). In other experiments, we induced activity levels in MB neurons throughout development until adult day 10 by expressing a heat-activated transient potential receptor cation channel, *dTrpA1*[55], in MBs of animals raised at 29 °C. In contrast to MB-specific autophagy attenuation, however, these flies displayed morphological defects in their MBs. A continuous MB-specific attenuation of synaptic release led to nearly full physical loss of MBs in *ok107/shi*[ts] and severely malformed MBs in *vt30559/shi*[ts] (Supplementary Fig. 12). A constitutive increase in neuronal activity of MB neurons, on the other hand, gave signs of axonal misrouting (Supplementary Fig. 12).

In compliance with developmental defects in *ok107/shi*[ts] and in *vt30559/shi*[ts], a lack of BRP staining was observed in the otherwise MB-region in *ok107/shi*[ts] and *vt30559/shi*[ts] (Supplementary Fig. 13). Contrary to MB-specific inhibition of autophagy, BRP staining levels in the remaining brain, however, were not increased but instead appeared rather even slightly reduced throughout the brain of *ok107/shi*[ts] and *vt30559/shi*[ts] flies (Supplementary Fig. 13). We also observed a strong reduction in synaptic BRP staining intensity in both *ok107/dTrpA1* and *vt30559/dTrpA1* (Supplementary Fig. 14). Here, BRP signal in the cell body region of KCs and in *Pars Intercerebralis* neurons was observed, regions normally devoid of BRP signal, potentially suggesting transport defects in these situations (Supplementary Fig. 14). Collectively, we conclude that a modulation of fast synaptic transmission should not be responsible for the spreading phenotypes observed after MB-specific KD of autophagy.

Apart from fast synaptic transmission, neuropeptides are major signals controlling nervous system functional adaptation and longer lasting plasticity processes[56–59]. NPF family neuropeptides are the dominant neuropeptide species expressed by the MB (Fig. 5a)[60].

Notably, in rodent hypothalamus, expression of NPF-family neuropeptide, NPY has been shown to intersect with the autophagic status of NPY secreting cells[61,62]. In fact, we noticed an about 40% reduction in sNPF transcript levels (relative fold change: $0.6417 \pm 0.1217$, $^*p < 0.05$, Paired *t*-test, $n = 6$) over whole dissected brains upon MB-specific attenuation of autophagy. Consistently, in immunostainings, the prominent MB expression of short NPF (sNPF) precursor peptide in the MB appeared about 50% reduced upon attenuation of autophagy in MBs (Fig. 5b–d). Outside the MBs, sNPF appeared somewhat reduced as well, though to a lesser but still significant degree (Fig. 5e–g). Notably, in aged animals, a very comparable reduction of sNPF peptide precursor levels was observed as well (Fig. 5h–q).

We then analyzed sNPF hypomorph flies (*sNPF*[c00448]), which by immunostainings displayed a ~50% decline in the levels of the sNPF peptide precursor in the MB (Fig. 6a–c), means an extent of reduction very similar to aged animals and as after MB-specific KD of autophagy. Brains of *sNPF*[c00448] flies showed significantly increased levels of synaptic BRP label over the brain as well (Fig. 6d–f). Notably, Neuropeptide F (NPF) family neuropeptides are suspected to counteract age-related phenotypes of the mammalian nervous system[63]. Given that we find sNPF levels to be similarly reduced in *atg5* and *atg9* KD, reduced sNPF signaling could well mediate effects of impaired MB-specific autophagy, provoking age-typical phenotypes.

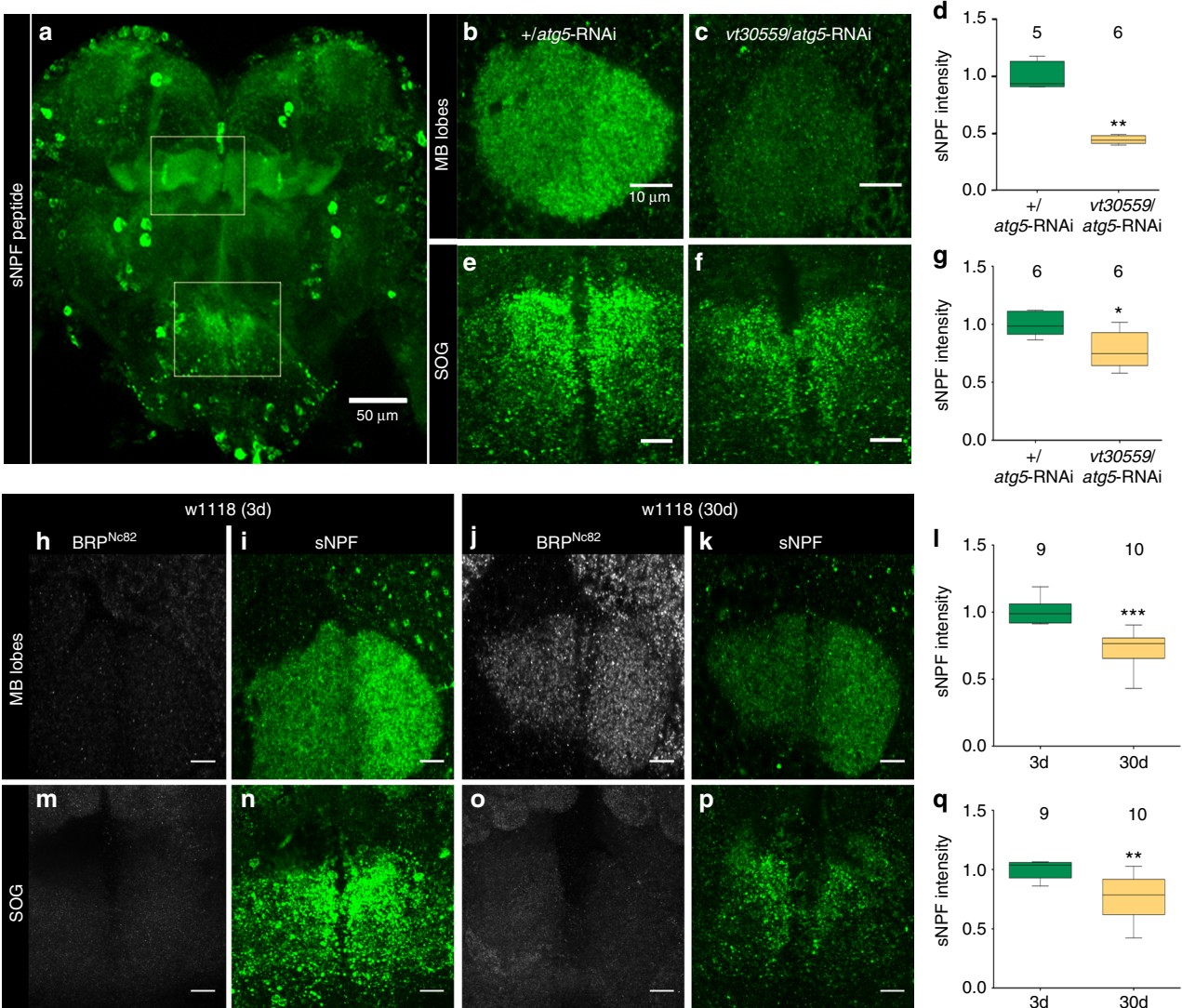

**Fig. 5** sNPF levels decline with mushroom body-specific attenuation of autophagy and with aging. **a** Adult brain immunostained for sNPF-precursor. Scale bar: 50 μm. **b** MB lobes of +/atg5-RNAi and **c** vt30559/atg5-RNAi immunostained for sNPF precursor (Single Z-plane). Scale bar: 10 μm. **d** Quantification of signal intensity of sNPF peptide precursor in the MB lobes normalized to control flies (5–6 independent animals; **$p < 0.01$; Mann–Whitney $U$-test). **e** Subesophageal ganglion (SOG) of +/atg5-RNAi and **f** vt30559/atg5-RNAi immunostained for sNPF precursor (Z-projection). Scale bar: 10 μm. **g** Quantification of signal intensity of sNPF-peptide precursor in SOG normalized to control flies ($n = 6$ independent animals; *$p < 0.05$; Mann–Whitney $U$-test). MB lobes of (**h**) 3-day-old w1118 and (**j**) 30-day-old w1118 immunostained for BRP[Nc82] (single Z-plane). Scale bar: 10 μm. MB lobes of (**i**) 3-day-old w1118 and (**k**) 30-day-old w1118 immunostained for sNPF (Single Z-plane). Scale bar: 10 μm. **l** Quantification of signal intensity of sNPF peptide precursor in the MB lobes normalized to control flies (9–10 independent animals; ***$p < 0.001$; Mann–Whitney $U$-test). SOG of (**m**) 3-day-old w1118 and (**o**) 30-day-old w1118 immunostained for BRP[Nc82] (Single Z-plane). Scale bar: 10 μm. SOG of (**n**) 3-day-old w1118 and (**p**) 30 days w1118 immunostained for sNPF (Z-projection). Scale bar: 10 μm. **q** Quantification of signal intensity of sNPF-peptide precursor in SOG normalized to control flies (9–10 independent animals; ***$p < 0.01$; Mann–Whitney $U$-test). In the box and whisker plots, the middle line of a box represents the median (50th percentile) and the terminal lines of a box represent the 25th and 75th percentile. The whiskers represent the lowest and the highest value

We next tested the role of MB sNPF signaling by targeting the sNPF-specific receptor (sNPFR). Notably, KD of *snpfr* using MB driver line *ok107*-Gal4 resulted in a very prominent brain-wide upregulation of synaptic BRP levels (Fig. 6g–i; also see Discussion). We also analyzed the ultrastructural status of AZs after MB-specific KD of the *snpfr*. We here analyzed PN presynapses within the Calyx (PNs are not part of the *ok107*-Gal4 expression domain). As expected from the BRP confocal staining experiments, a metaplastic increase of AZ ultrastructural sizes was observed (Fig. 6j–l).

We finally tested whether a reduction in MB-specific sNPF signaling and the resulting change of synaptic status would, as to be expected, affect the ASM component of memory specifically.

This analysis is complicated given that genuine smell scores were, as expected[64], affected by reduced sNPF signaling (Supplementary Fig. 15). Still, we found that ASM but not ARM scores were significantly reduced in *snpf* hypomorphs and also after MB-specific KD of *snpfr* (Supplementary Fig. 15). Given that MB-specific KD of the sNPF-receptor was obviously sufficient to provoke the brain-wide metaplastic change, as well as to prematurely impair normally age-sensitive memory, we conclude that autocrine signaling of Neuropeptide F in MB fulfills the criteria for a neuropeptide signal to keep presynaptic metaplasticity in a competent state required for memory formation. Since attenuation of autophagy within the MB obviously reduces sNPF ligand levels, the spreading effects of autophagy onto memory

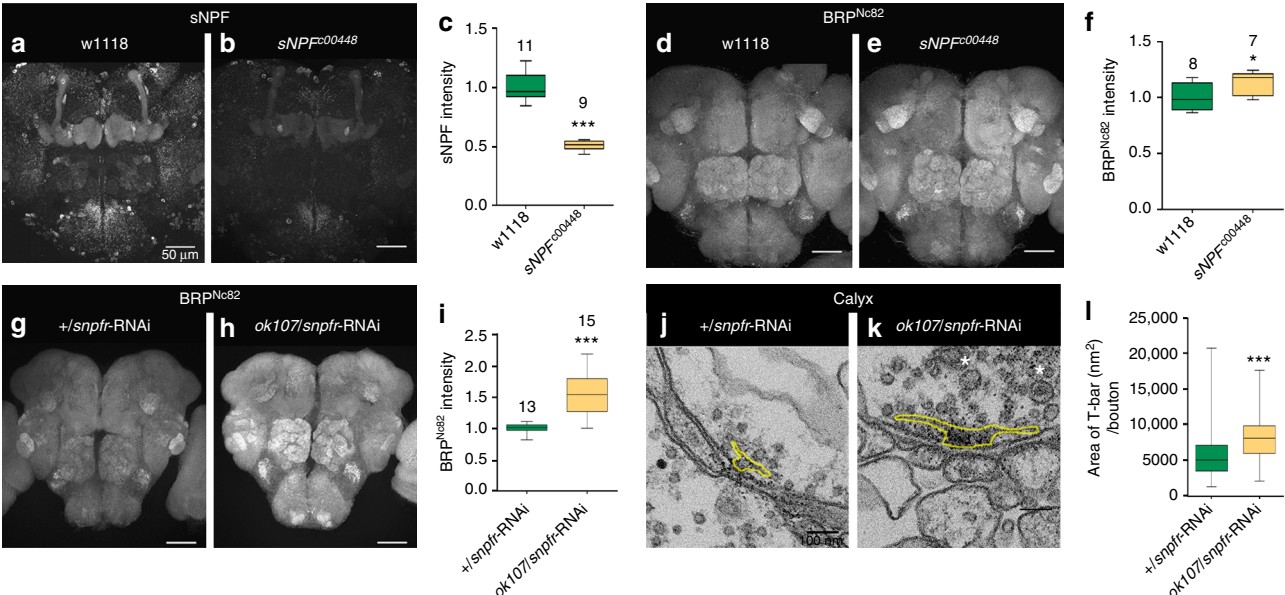

**Fig. 6** sNPF signaling connects autophagy suppression with presynaptic metaplasticity. Adult brain, 5 days, (**a**) w1118 and (**b**) *sNPF^c00448* immunostained for sNPF peptide precursor. Scale bar: 50 μm. **c** Quantification of sNPF peptide precursor within the central brain region normalized to control flies. (*n* = 9–11 independent brains; ***p* < 0.001; Mann–Whitney *U*-test). Adult brain, 5 days, (**d**) w1118 and (**e**) *sNPF^c00448* flies, immunostained for BRP^Nc82. Scale bar: 50 μm. **f** Quantification of BRP^Nc82 within the central brain region normalized to control flies (*n* = 7–8 independent brains; **p* < 0.05; Mann–Whitney *U*-test). Adult brain, 5 days, (**g**) +/*snpfr*-RNAi and (**h**) *ok107*/*snpfr*-RNAi flies, immunostained for BRP^Nc82. Scale bar: 50 μm. **i** Quantification of BRP^Nc82 within the central brain region normalized to control flies (*n* = 13–15 independent brains; ***p* < 0.001; Mann–Whitney *U*-test). Electron micrograph showing presynaptic specializations at PN-to-KC synapses in Calyx of (**j**) +/*snpfr*-RNAi and (**k**) *ok107*/*snpfr*-RNAi animals. Scale bar: 100 nm. **l** Quantification representing the average T-bar size in +/*snpfr*-RNAi and *ok107*/*snpfr*-RNAi animals (*n* = 206 electron micrographs across eight independent animals, with at least 20 T-bars per animal; ***p* < 0.001; Mann–Whitney *U*-test). In the box and whisker plots, the middle line of a box represents the median (50th percentile) and the terminal lines of a box represent the 25th and 75th percentile. The whiskers represent the lowest and the highest value

aversive brain-wide metaplasticity and consequently age-sensitive memory formation might involve this particular neuropeptide, which in rodents links autophagy and metabolic state with neuronal function and plasticity[65,66].

## Discussion

The maintenance of neuronal homeostasis is severely threatened by aging. The strictly postnatal character of deficits observed after KD of core autophagy machinery triggered the hope that autophagy might have a specific relation to the aging process[67]. The last few years have indeed seen an accumulation of evidences that the efficiency of autophagic clearance in neurons declines with age on organismal level[4,37,68–70]. Hence, rejuvenating autophagy in aging neurons is considered a promising strategy to restore cognitive performance. Successfully exploring this direction will, however, depend on deepening our insights at the intersection of autophagy, the relevant neuronal sub-cellular compartments, importantly synaptic specializations, and relevant neuron populations/brain regions.

The endogenous polyamine spermidine has prominent cardioprotective and neuro-protective effects[71] and recent work finds spermidine restoration to counteract otherwise deteriorating health in aging mice in an autophagy-dependent manner[72]. In *Drosophila*, restoring spermidine specifically suppressed age-induced decay in their ability to form olfactory memories, again in an autophagy-dependent manner[37]. Concomitantly, in the aged *Drosophila* brain, we previously found a brain-wide, age-induced upshift in the ultrastructural size (EM: larger T-bars; STED: increased diameter of BRP scaffold) of presynaptic AZs (metaplasticity). Two findings causally linked this upshift to

decreased olfactory memory performance. First, when continuously fed with spermidine, flies of 30 days of age (normally suffering from a complete loss of age-sensitive component of memory) were largely protected from these changes. Secondly, genetically provoking this up-shift eliminated the normally age-sensitive memory component in young animals already[39]. An upshift in the AZ size should increase synaptic strength[52], evident in increased SV release in response to natural odors observed in aged but not aged-spermidine-fed flies[39]. Presynaptic plasticity is crucial for forming memory traces in *Drosophila*[73]. Our previous work thus suggests that this presynaptic metaplasticity shifts the operational range of synapses in a way that they become unable to execute the plastic changes faithfully in response to conditioning stimuli.

We here further addressed the relation between defective autophagy, presynaptic ultrastructure and plasticity and olfactory memory formation. Autophagosome biogenesis is very dominant close to presynaptic specializations in distal axons in compartmentalized fashion and efficient macro-autophagy is essential for neuronal homeostasis and survival[7,8]. Retrograde transport of autophagosomes might play a role in broader neuronal signaling processes, promoting neuronal complexity and preventing neurodegeneration. Surprisingly, however, our data do not favor a direct substrate relationship between AZ proteins and autophagy. Instead, we find evidence for a seemingly non-cell autonomous relation between brain-wide synapse organization and the autophagic status of the mere MB. After genetic impairment of autophagy (via *atg5* or *atg9* KD) using two different MB-specific Gal4-driver lines, we observed the presynaptic metaplasticity across the *Drosophila* olfactory system and beyond. While the autophagic arrest (p62 staining) was largely limited to the

expression domain of these drivers, the synapses were pushed towards a state of metaplasticity. Since the ultrastructural size of AZs and the per AZ BRP levels[39] increased equally in aged and MB-autophagy-challenged animals, we conclude that the autophagic status of the MB neuron population executes a signaling process, which can control the per AZ amounts of BRP and other AZ proteins. Further studies are warranted to dissect the nature of these signaling processes.

Notably, accumulating evidences support the important role of neuropeptide Y (NPY) in aging and lifespan determination[63]. NPY levels decrease with age in mice and re-substituting NPY is able to counteract age-induced changes of the brain at several levels[63]. A cross-talk between autophagy and NPY in regulating the feeding behavior has been demonstrated in mice[61,62].

We here found that transcript expression level of an NPY family member (sNPF) are controlled by autophagy within the MBs. We used an *snpf hypomorph* allele mimicking the MB reduction of sNPF of the MB-specific autophagy KD situations as well as the sNPF expression in aged animals. In this hypomorph allele we observed a similar up regulation in BRP Nc82 signal. KD of the *snpfr* using an MB-specific driver drove the brain-wide metaplastic change even stronger than the sNPF hypomorph (obviously only partially affecting the sNPF-specific signaling). This scenario in ultrastructural detail resembled both the age-induced and MB-specific autophagy-KD-induced metaplasticity phenotypes. These results, therefore, support the essential role of MB in integrating the metabolic state of *Drosophila* in an autocrine fashion to modulate the presynaptic release scaffold state throughout the fly brain. The mechanistic basis of this exciting regulation warrants further investigation. Interestingly, elevated cAMP signaling is generally driving plasticity in *Drosophila* neurons, while sNPF signaling is meant to reduce cAMP[74] and thus potentially might be able to reset plastic changes such as increased BRP levels. In apparent contradiction to sNPF signaling directly widely controlling metaplasticity is our finding that MB-specific KD of the sNPFR sufficed to increase BRP levels. At this moment, we can only speculate as to why KD of sNPF-receptor also results in extended metaplastic changes. Potentially, sNPF-receptor signaling within the MB might be important to control sNPF secretion in a physiological manner via a quasi-autocrine mechanism.

Intriguingly, the metaplastic state characterized both aged and MB-specific autophagy KD animals, and in both cases provoked a specific loss of the ASM component of memory. Notably, olfactory MTM measured here, are considered to be the direct precursor of olfactory LTM, which in turn have been shown to be energetically costly[31]. Notably, autophagy and NPY signaling are prime candidate mechanisms for the therapy of age-induced cognitive processes[63,71,75].

Recent research has uncovered several examples connecting autophagy and hormonal-type regulations interacting between organ systems in non-cell autonomous regimes. For instance, Atg18 acts *non*-cell autonomously both in neurons and in intestines to firstly, maintain the wild-type lifespan of *C.elegans* and secondly, to respond to the dietary restriction and DAF-2 longevity signals[44]. Atg18 in chemosensory neurons and intestines acts in parallel and converges on unidentified neurons that secrete neuropeptides to mediate the influence of Daf-2 on *C. elegans* lifespan through the transcription factor DAF-16/FOXO in response to reduced IGF signaling[44]. In *Drosophila*, neuronal up-regulation of AMPK induces autophagy, via up-regulation of Atg1 *non*-cell autonomously in intestines and slows intestinal aging and vice versa. Moreover, up-regulation of Atg1 in neurons extends lifespan and maintains intestinal homeostasis during aging and these inter-tissue effects of AMPK/Atg1 were linked to altered insulin-like signaling[43]. On the contrary, we found the

insulin producing cells (IPCs) themselves to not mediate the observed metaplastic state, as neither the KD of *atg9* nor the KD of *snpfr* in *Pars intercerebralis* had any impact on the synaptic status of these flies.

Autophagy regulation is tightly connected to cellular energetics, nutrient recycling, and the maintenance of cellular energy status[76]. The fruit fly can evaluate its metabolic state by integrating hunger and satiety signals at the very KC-to-MBON synapses in MB under control of dopaminergic neurons to control hunger-driven food-seeking behavior[77]. At the same time, long-term memory encoding necessitates an increase in MB energy flux with dopamine signaling mediating this energy switch in the MB[31]. In line with these findings, we here now provide a modeling basis to study these delicate relations in an exemplary fashion. Taken together, our data suggest that MB integrates the metabolic state of the flies via cross talk between autophagy and sNPF signaling with the decision whether to form memories or not and a block in this cross talk with aging gives rise to synaptic metaplasticity which initiates the age-induced memory impairment in *Drosophila*. It is tempting to speculate that the MB executes hierarchically, a high-level control integrating the metabolic and caloric situation with a life-strategy decision of whether or not to form mid-term memories.

## Methods

**Fly rearing**. Fly-strains were reared under standard laboratory conditions at 25 °C and 65% humidity with constant 12 h:12 h light:dark cycle, unless otherwise stated. Flies were raised on standard fly food (adapted from https://bdsc.indiana.edu/information/recipes/bloomfood.html with minor modifications). For KD studies, flies were mated at 25 °C and the F1 progeny were allowed to develop and age (until desired age) at 29 °C. Flies used in all experiments are F1 progeny. For aging, flies were collected once every 2 days (preferably evening) and flipped every 2–3 days on fresh food until desired age was reached.

Isogenized w[1118] strain were used as the wild-type control. *snpfr*-RNAi was kindly provided by Prof. Dr. Ilona Grunwald Kadow (Munich Center for Neuroscience). *sNPF[c00448]* flies were kindly provided by Dr. Peter Soba (Universitätsklinikum Hamburg-Eppendorf). *dilp2*-Gal4 (#37516), *atg7*-RNAi (#27707, #34369), *atg5*-RNAi (#34899, #27551), *atg9*-RNAi (#34901), *atg8*-RNAi (#28989), *syx17*-RNAi (#25896) were obtained from the Bloomington *Drosophila* stock center. *vt30559*-Gal4 (#206077) and *atg17*-RNAi (#KK104864) were obtained from Vienna *Drosophila* Resource Center. In addition, *elav*-Gal4, *appl*-Gal4, *gh146*-Gal4, *ok107*-Gal4, *ok107*-Gal4; *mb247*-Gal80 and *ok107*-Gal4; *tub*-Gal80[ts] were used.

**Quantitative real-time PCR**. For validation of RNA KD efficiency of *atg5* and *atg9* and to quantify differences in transcription of *snpf*, qRT-PCR was performed. Total RNA was isolated from whole brains of 50-day-old, 10-day-old female flies using TRIzol reagent (Invitrogen). RNA concentration was measured using spectrophotometer (NanoDrop) and 500 ng of RNA was converted to cDNA using the SuperScript III First Strand Synthesis System (Invitrogen) according to the manufacturer's instructions. The primers used to amplify the *atg5* were as follows: *atg5-Forward* (GCACTACATGTCCTGCCTGA) and *atg5-Reverse* (AGATTCGCAGGGGAATGTTT). The primers used to amplify the *snpf* were as follows: *snpf-Forward* (CAAAAAGCGTGGCATACATT) and *snpf-Reverse* (AATGTCCGGATTTCAAGGAG). The primers used to amplify the *atg9* were as follows: *atg9-Forward* (TTGTCCAGATCCGAATCCTC) and *atg9-Reverse* (TCGTCTGGCTACTTGCCTTTT). *actin5c* was used as a reference gene for normalization and calculation of fold change differences between control and experimental group(s). The primers used to amplify *actin5c* were as follows: *actin5c-Forward* (TTGTCTGGGCAAGAGGATCAG) and *actin5c-Reverse* (ACCACTCGGCACTTGCACTTTC). All primers were tested for their amplification efficiency according to standard methods. qRT-PCR was performed using the Dynamo Flash SYBR green master mix (Thermo-Fischer # F415L) and the Agilent Technologies Stratagene Mx3005P Real-time PCR system according to the manufacturer's instructions. The threshold cycle (Ct) is the point where each kinetic curve reaches a common arbitrary fluorescence level (AFL), placed to intersect each curve in the region of exponential increase. Subsequently, the Ct values of experimental group were subtracted from that of control group, resulting in $-\Delta Ct$ and the fold change was calculated as $2^{-\Delta\Delta Ct}$. Values are presented as mean ± SE of triplicate assay.

**Protein extraction, SDS–PAGE, and western analysis**. To detect p62 and Atg8a, five female fly brains were homogenized in 50 µl 2% SDS buffer containing protease inhibitors. An amount equivalent to 1 brain was loaded and resolved on 4–20% gradient gels (#4561096; BioRad), followed by electroblotting to nitrocellulose

membranes (#10401396; Millipore). Subsequently, blots were probed with antibodies against Tubulin (loading control), p62 and Atg8a (see antibodies for further information). Immunoblots were scanned and analyzed using ImageJ software. The relative amounts of p62 and Atg8a proteins from individual samples were corrected using antibody to Tubulin as loading control.

**Antibodies**. The following antibody dilutions were used for Confocal microscopy: MαBRP$^{Nc82}$ (1:25; DSHB), MαFasII$^{1D4}$ (1:40; DSHB) Rbαp62 (1:2000; Gabor Juhasz), RbαsNPF (1:2000; Jan Veenstra), GαM Cy3 (1:500; Ab97035), and GαRb Alexa 488(1:500; A11008), MαGFP (1:1000; Abcam, A11120), RbαAnnexinV (1:100, Ab14196), RbαDcp-1 (1:100, Asp216).

The following antibody dilutions were used for super-resolution STED microscopy: MαBRP$^{Nc82}$ (1:20; DSHB), Rbαp62 (1:1000; Gabor Juhasz), RbαsNPF (1:1000; Jan Veenstra), GαM aberrior star 635p(1:200; #200020075), and GαRb Alexa 594 (1:200; A11037).

The following antibody dilutions were used for Western blots: Rbαp62 (1:2500; Gabor Juhasz), MαTubulin (1:10,000; T9026), RbαAtg8a (1:1000; Ab109364), GαM Peroxidase (1:5000; Dianova 115035166), and GαRb Peroxidase (1:5000; Dianova 111035144).

**Adult *Drosophila* brain dissection and immunohistochemistry**. The adult brain dissections were always done between 8 a.m. and 11 a.m. Adult brains were dissected in ice-cold hemolymph-like saline (HL3) solution and immediately, fixed in 4% paraformaldehyde at room temperature (RT) for 30 min. After fixation the brains were incubated with PBS containing 0.5% Triton X-100 (0.5% PBT) for 30 min. Afterwards they were blocked in 10% normal goat serum (NGS; v/v) for 2 h at RT. For primary antibody treatment, samples were incubated in 0.5% PBT containing 5% NGS, 0.1% sodium azide, and primary antibodies for 48 h at 4 °C. After primary antibody incubation, brains were washed in 0.5% PBT for 6 × 30 min at RT. All samples were then incubated in 0.5% PBT with 5% NGS, 0.1% sodium azide containing the secondary antibodies for 24 h at 4 °C. Brains were washed in 0.5% PBT for 6 × 30 min at RT followed by overnight incubation in Vectashield® (Vector Laboratories) before confocal scanning. The dilutions for various antibodies used for immunohistochemistry are mentioned in Antibodies section.

**Image acquisition**. Conventional confocal images were acquired with a Leica TCS SP8 confocal microscope (Leica Microsystems) using a ×20, 0.7 NA oil objective for whole-brain imaging. All images were acquired using Leica LAS X software. Lateral pixel size was set to values around 300 nm. Exact values varied depending on situation. Typically 1024 × 1024 images were scanned at 600 Hz using 4x line averaging.

STED microscopy was performed using Leica Microsystems TCS SP8 gSTED 3x set-up equipped with pulsed white light laser (WLL; ~80 ps pulse width, 80-MHz repetition rate; NKT Photonics) and two STED lasers for depletion (continuous wave at 592 nm, pulsed at 775 nm). The pulsed 775 nm STED laser was triggered by the WLL. Images were acquired with ×100, 1.4NA oil immersion objective. 1024 × 1024 pixel resolution STED images were scanned at 600 Hz using 8x line averaging. Lateral pixel size was set to values ~18 nm with z-stack of three images, step size 0.13 nm for better PSF estimation. To minimize thermal drift, the microscope was housed in an incubation chamber. STED images were processed using the Huygens deconvolution software (SVI, the Netherlands). STED images were acquired at the cellular imaging facility of the Leibniz Instititute for Molecular Pharmacology Berlin, Germany.

**Image processing and analysis**. Segmentation of 3D image stacks of the central body region of brains was done using Amira® software, Visage Imaging GmbH. The first step was to separate the object of interest (central brain region) from the background (part of optical lobes on both hemispheres). A unique label was defined for each region in the first fluorescence channel (e.g. Nc82). This was done by manually assigning the central brain region to interior regions on the basis of the voxel values (volumetric pixels). By this procedure, each voxel value outside the central brain region was excluded from the interior label (i.e. the area belonging to the central brain region of each focal plane was included for later measurements). A full statistical analysis of the image data associated with the segmented materials was obtained by applying Material Statistics module of the Amira® software, in which the mean gray value of the interior region (central brain region) is calculated. The median voxel values of the central brain regions were compared, as measured in individual adult brains, in order to evaluate the synaptic marker label.

In case of p62/Ref(2)p and sNPF peptide precursor, images were quantified using FIJI software (http://fiji.sc/FIJI). Confocal stacks were merged into a single z-plane by using the maximum projection function. Subsequently, the region of central brain was manually selected (using free-hand function) and absolute fluorescence intensity was measured and normalized to the area of the central brain for each brain.

For STED analysis, deconvolved BRP spots (stained with monoclonal Nc82 antibody) were processed in ImageJ. The diameters of planar oriented BRP rings were measured using the line tool of ImageJ. The distance from intensity maximum to intensity maximum was acquired in the plot window of individual hand-drawn lines and transferred to Microsoft excel.

**Cell counting**. For cell counting, we collected confocal stacks at 0.5 µm intervals with a ×63 objective lens. The posterior region of the MB was zoomed at ×1.5 magnification so that all the Gal4 expressing Kenyon cell bodies are in a frame. The posterior MBs in the left and right hemispheres were separately scanned and analyzed. For the quantitative analysis, brains were scanned with comparable intensity and offset. Images of the confocal stacks were analyzed with the open-source softwareFiJi. Randomly chosen stacks with non-overlapping cell bodies were examined manually to quantify GFP-positive cell bodies.

**Electron microscopy**. Brains were dissected in HL3 solution and fixed for 20 min at RT with 4% paraformaldehyde and 0.5% glutaraldehyde in PBS. Subsequently, the brains were incubated overnight at 4 °C with 2% glutaraldehyde in buffer containing 0.1 M sodium cacodylate at pH 7.2. Brains were then washed 3× in cacodylate buffer for 10 min at 20–30 °C. Afterwards, the brains were incubated with 1% Osmium tetroxide and 0.8% KFeCn (in 0.1 M cacodylate buffer) for 90 min on ice. Brains were then washed with cacodylate buffer for 10 min on ice and then three quick washed with distilled water. The brains were stained with 1% uranylacetate (w/v) for 90 min on ice and dehydrated through a series of increasing alcohol concentrations. Samples were embedded in EPON resin by incubation sequentially in ethanol/EPON 1:1 solution for 45 and 90 min at 20–30 °C, then in pure EPON overnight at 15–20 °C. Thereafter, the resin was changed once and brains were embedded in a single block at 60 °C to allow for polymerization of the resin.

After embedding, sections of 60 nm each were cut using a Leica Ultracut E ultramicrotome equipped with a 2 mm diamond knife. Sections were collected on 100 mesh copper grids (Plano GmbH, Germany) coated with 0.1% Pioloform resin. Contrast was enhanced by placing the grids in 2% uranyl acetate for 2 min, followed by washing with water three times and then incubation in lead citrate for 1.5 min. The grids were washed 3× with water and dried. Images were acquired fully automatically on a FEI tecnai Spirit transmission electron microscope operated at 120 kV equipped with a FEI 2 K Eagle CCD camera using Leginon. Regions of interest were first selected at ×560 nominal magnification and then successively imaged at ×4400, ×11,000 and ×21,000 nominal magnification, respectively.

**Aversive olfactory conditioning in flies**. Behavioral experiments were performed in dim red light in 25 °C and ~70% humidity with 3-octanol and 4-methylcyclohexanol (1:100 dilution in mineral oil presented in a 14 mm cup for both odors) serving as olfactory cues and 120 V alternate current served as behavioral reinforce. Standard single-cycle olfactory associative memory was performed as previously described with minor modifications. Briefly 80–100 flies received one training session, during which they were exposed sequentially to one odor (conditioned stimulus, CS$^+$; 3-octanol or 4-methylcyclohexanol) paired with electric shock (unconditioned stimulus, US) followed by a rest of 30 s and then to a second odor (CS$^-$; 4-methylcyclohexanol or 3-octanol) without US for 60 s. During testing, flies were exposed simultaneously to the CS$^+$ and CS$^-$ in a T-maze for 30 s.

For short-term memory (STM; memory tested immediately after odor conditioning), the conditioned odor avoidance was tested immediately after training. Subsequently, flies trapped in either T-maze arm were anaesthetized and counted. For each distribution, a performance index (PI) was calculated as

$$PI = \frac{CS^- - CS^+}{CS^- - CS^+} x100 \tag{1}$$

A 50:50 distribution (no learning) yields a PI of 0 and a 0:100 distribution away from the CS$^+$ gives a PI of 100. PI was calculated for both reciprocal indices for the two odors and a final PI was calculated as mean of PI$_I$ and PI$_{II}$.

For MTM, flies were trained as explained above, but tested 1 h after training. For separation of consolidated ARM and labile ASM, the flies were trained and one group was cooled in an ice-bath for 90 s, 30 min after training. The flies were allowed a recovery period of 30 min, i.e. 1 h after training onset. Since labile ASM is erased by this procedure, performance of the cooled group is solely due to ARM. In other words, ASM was calculated by subtracting the PI of ARM from that of median MTM.

**Statistics**. Unless otherwise stated, the data was analyzed wth GraphPad Prism 5 software. Asterisks are used to indicate statistical significance of the results (*$p <$ 0.05; **$p < 0.01$; ***$p < 0.001$; $^{ns}p > 0.5$). No statistical methods were used to pre-determine sample sizes but our sample sizes are similar to those reported in previous publications. Wherever possible, data were collected with the investigator blind to the genotypes, treatment, and age of genotypes. The data collection and data processing were done in parallel in randomized order for all experiments. Two groups were compared using the non-parametric Mann–Whitney $U$-tests while more than two groups were compared using one-way ANOVA with different post-hoc correction or Kruskal–Wallis test, which have been mentioned in the figure legends. Number of independent experiments, '$n$'s are mentioned in figure legends.

This study did not use animals and/or animal-derived materials for which ethical approval is required.

**Reporting summary**. Further information on experimental design is available in the Nature Research Reporting Summary linked to this article.

## Data availability

The data that supports the findings of this study are available from the corresponding author (SJS) upon reasonable request.

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

## Acknowledgements

We would like to thank Dr. Peter Soba (Universitätsklinikum Hamburg-Eppendorf), Prof. Dr. Ilona Grunwald Kadow (Munich Center for Neurosciences), the Bloomington *Drosophila* Stock Center and Vienna *Drosophila* Research Center for fly stocks. We are grateful to Jan Veenstra for the sNPF peptide precursor antibody. We would like to acknowledge the assistance of the Core Facility BioSupraMol supported by the DFG. We would like to thank Christine Quentin for her excellent technical assistance. We would like to thank Prof. Florian Heyd for their RT-PCR set-up. We would like to thank Atefeh Pooryasin for providing *ok107*-Gal4;*mb247*-Gal80 and *ok107*-Gal4;*tub*-Gal80ts flies. This work was supported by grants from the Bundesministerium für Bildung and Forschung (SMARTAGE, 01GQ1420A to S.J.S. and C.B.B., the Deutsche For-schungsgemeinschaft to S.J.S. and A.B. (Exc 257, TP A3, and A6 SFB 958; SFB 740 TP C09; SFB1315). F.M. is grateful to the Austrian Science Fund FWF for grants P23490-B12, P24381, P27893, I1000, and grant 'SFB Lipotox' and to BMWFW and the Karl-Franzens University for grant 'Unkonventionelle Forschung'. The funders had no role in study design, data collection and analysis, decision to publish or preparation of the manuscript. Stephan Sigrist is indebted to the DZNE (Deutsches Zentrum für neuro-degenerative Erkrankungen) and FOR 2705.

## Author contributions

A.B. and S.J.S. conceived the project. A.B. and C.B.B. performed *Drosophila* genetics. A.B. performed the confocal imaging experiments and evaluated the data. A.B., M.M. and M. L. performed the STED imaging experiments and A.B. analyzed the data. A.B. analyzed the EM data. C.B.B. performed the behavioral experiments and C.B.B. and A.B. analyzed the data. A.B. performed qRT-PCR and Western blots and analyzed subsequent data. F.M., G.J. and S.J.S. provided the necessary resources. A.B., F.M. and S.J.S. wrote the manuscript with input from all the authors.

## Additional information

**Competing interests:** The authors declare no competing interests.

**Journal Peer Review Information**: *Nature Communications* thanks the anonymous reviewer(s) for their contribution to the peer review of this work. Peer reviewer reports are available.

