## [Peer Review File · Nature Communications]

Reviewers' comments:

Reviewer #1 (Remarks to the Author):

The authors describe an intriguing mechanism of non-cell autonomous presynaptic organization changes in the brain as a result of manipulating autophagy in the mushroom bodies. They also show these changes are the result of autophagy-controlled defects in NPY release from the mushroom bodies. Hence, autophagy in specific brain centers controls (tunes) the information processing in the rest of the brain. This is exciting work that will open new directions of research.

Specific questions/concerns:

1-In figure 2c and d the authors state that MTM is already reduced at 10days when the Atg proteins are knocked down whereas in wild type flies lower MTM only occurs in older flies. For the sake of comparison, it would be useful to include one such later time point, showing that controls now also have lower MTM, similar to the knock down conditions at 10d (and at that later timepoint). A similar comparison between 10d old and older animals would be useful for ARM and ASM as well.

2-In Fig 3 the authors assess BRP levels in the entire brain upon knock down of autophagy genes only in MBs. They find BRP is upregulated everywhere. I suppose Refp2 is not upregulated everywhere? There is some data in the supplement on this, but there is no co-labeling of Refp2 with a marker of the cells that express Gal4 (and RNAi to autophagy genes).

3-In fig 3 the authors block the knock down of Atg5 using Gal80. This is a good control experiment, but an even cooler experiment would be to knock down Atg5 in the entire brain and prevent knock down only in the mushroom bodies. If what the authors propose is correct, this too should rescue the phenotypes (memory and BRP labeling).

4-There is less expression of sNPF when autophagy is inhibited. Is there also less RNA expression? How do the authors envisage that lower autophagy causes less sNPF to be expressed? Can this be discussed?

5-Did the authors try to over express sNPF in Kenyon cells where autophagy genes were knocked down to assess if this rescues metaplasticity?

6-I lacked a discussion on how the authors think sNPF results in the regulation of BRP levels?

7-The quality of the western blot in fig 1g is not optimal...

8-What are the efficiencies of RNA knock down throughout the paper?

9-Check references to figures because some are not correctly referring to the right panels

Reviewer #2 (Remarks to the Author):

Bhukel, Beuschel, Maglione, Junhasz, Madeo and Sigrist | Manuscript submitted to Nature Communications

"Autophagy within the mushroom body learning center protects from synapse aging in a non-cell autonomous manner"

The mushroom body is a brain center required for the formation and storage of associative memories.

In their manuscript, Bhukel et al. report evidence suggesting that a reduction of autophagy in the mushroom body impairs the formation of associative memories. The evidence supporting this conclusion is twofold: first, the authors found that brain-wide and mushroom body specific attenuation of autophagy — which was carried out using the GAL4 system and RNA interference — leads to memory defects; second, the authors observed that attenuation of autophagy in the mushroom body leads to brain-wide increase in the number of presynaptic active zones. They refer to this increase as 'synaptic metaplasticity'. The authors also suggest that autophagy is necessary to maintain synaptic homeostasis in aging brains and that neuropeptide F — a neuropeptide secreted by the mushroom body — might be a potential non-cell autonomous mediator of this synaptic metaplasticity. The evidence supporting this conclusion includes a comparative study with an hypomorph mutant of neuropeptide F and a knockdown of its receptor.

This manuscript proposes an interesting mechanism whereby neuropeptide F could be a key regulator of synaptic homeostasis in the fly brain. However, the experimental evidence supporting the claims required to support such a mechanism is weak and not as thorough as it should and could be. Mainly for this reason, I do not support the publication of this manuscript in Nature Communication. I think the manuscript needs several more experiments and is perhaps better fitted for a more specialized journal. Please find below a more detail criticism of the experiments and the conclusions reached by the authors.

Major points of criticism.

1. Overall effects of the manipulations used to block autophagy

To block autophagy, the authors used two transgenic lines that each contains a RNAi construct targeting genes important for autophagy, namely *atg5* and *atg9*. The authors used these tools in combination with the GAL4 expression system to reduce autophagy in the whole brain or in specific subpopulations of neurons. They gaged the effects of these manipulations by measuring the cellular levels of the ubiquitin binding scaffold protein p62, which accumulates when autophagy is blocked. The authors could use more controls than a simple p62 readout to determine how specific the observed defects are. For instance, are the signaling pathways induced during phagocytosis increased as well and are these changes specific to the tissues in which autophagy was manipulated? Are these manipulations causing side effects such as apoptosis? Strong experimental evidence is needed here as this technique — the manipulation of autophagy using these two RNAi constructs — is the technique most experiments are built on and the very foundation of this study.

2. Lack of explanation for Table 1 and Table 2

There is no legend for the Table 1 and Table 2 which makes it difficult to interpret the data reported these tables. Table 1 is useless and I cannot interpret the results reported in Table 2. The innate avoidance scores should be normalized and explained.

3. Subtlety of the learning defects and how they relate to defects seen in aging brains

The effects on short-term and mid-term learning, most of which are reported in Figure 2, are very subtle although statistically significant. Namely, there is a lot of residual learning seen in all manipulated animals. In light of the subtlety of these defects, it would have been worthwhile to explore the learning defects better and more carefully. Unfortunately, mass training of population of flies reported as a preference index is a very limited analysis. Because the learning deficits caused by the attenuation of autophagy in the mushroom body is the major finding made by this study, the experimental evidence supporting this claim needs to be stronger. Also, the authors mentioned that "KD of a core autophagy component within the MBs suffices to mimic the usual age-induced specific decay of the ASM component" but they did not compare the learning defects they observed to the ones normally seen in aged flies. This criticism applies to all the other experiments as well.

4. Changes in the number of presynaptic zones

The authors claim observing an increase in the number presynaptic zones — as measured by immunostaining using an antibody against bruchpilot, a master regulator of presynaptic zones — across the brain when autophagy is abolished only in the mushroom body. They refer to this phenomenon as 'presynaptic metaplasticity'. The evidence supporting this conclusion is rather weak. The comparison of the overall fluorescence of whole brains are provided as the major piece of evidence. Immunostaining on whole brains can be extremely variable and a more detailed study would be helpful to make this evidence stronger. Other evidence includes electron micrographs, which also show very subtle defects. The authors need stronger and more detailed evidence — including functional evidence — to support the claim that abolishing autophagy in the mushroom body affects the number of presynaptic zones across the entire brain.

5. Fast synaptic transmission is not responsible for the observed synaptic metaplasticity

The data supporting the claims made in lines 259 to 272 — that synaptic metaplasticity is not a consequence of increased or decreased neuronal activity in the mushroom body — should be provided as a supplemental figure.

5. Neuropeptide F as a regulator of 'presynaptic metaplasticity'

Lastly, the authors report evidence showing that blocking autophagy reduces the production of neuropeptide F made by the mushroom body. The authors also found that an hypomorph mutant of neuropeptide F and a mushroom-body specific knockdown of the neuropeptide F receptor both show an increase in synaptic metaplasticity, similarly as observed in the brain of flies in which autophagy was blocked in the mushroom body only. Why the knockdown of the receptor in the mushroom body only is sufficient to see these effects — and not in all neurons — is not discussed. The authors suggest, but do not show, that neuropeptide F is important to maintain presynaptic metaplasticity in aging brain, namely by maintaining homeostasis in the number of presynaptic zones. They suggest that a reduction in neuropeptide F might occur in older animals and be the cause age related learning defects. This last claim is very confusing because not properly explained as well as not properly tested, not directly tested.

Response prepared for Reviewer #1 (Remarks to the Author):

The authors describe an intriguing mechanism of non-cell autonomous presynaptic organization changes in the brain as a result of manipulating autophagy in the mushroom bodies. They also show these changes are the result of autophagy-controlled defects in NPY release from the mushroom bodies. Hence, autophagy in specific brain centers controls (tunes) the information processing in the rest of the brain. This is exciting work that will open new directions of research.

- We sincerely thank the Reviewer #1 for appreciating our work.

Specific questions:

1. In figure 2c and d the authors state that MTM is already reduced at 10days when the Atg proteins are knocked down whereas in wild type flies lower MTM only occurs in older flies. For the sake of comparison, it would be useful to include one such later time point, showing that controls now also have lower MTM, similar to the knock down conditions at 10d (and at that later timepoint). A similar comparison between 10d old and older animals would be useful for ARM and ASM as well.

- We thank the reviewer for having put our attention on this point. We previously showed that aged flies display reduced MTM scores tested 3 hours after training (Gupta et al., 2013). We now performed an additional experiment where we directly compared MTM scores of 10d control (+*atg5*-RNAi), 30d control and 10d KD animals (*elav/atg5*-RNAi) tested 1 hour after training. Indeed, the 1h-MTM scores of 10d KD animals were very comparable to the values of 30d control animals (Fig S3a). Notably, *elav/atg5*-RNAi animals did not survive in sufficient numbers to allow for testing 1h MTM at 30 days of age.

From our previous results, we can be sure that it is the ASM component which is sensitive to aging (means at 30 days) under our conditions (Gupta et al., 2013). We now directly compared ARM and ASM scores of 10d control (+*atg5*-RNAi), 30d control and 10d KD animals (*elav/atg5*-RNAi) tested 1 hour after training. While ARM scores stay comparable among genotypes at respective age(s), the KD of autophagy with *elav*-Gal4 specifically affects the ASM at 10 days (Fig S3, b-c) and autophagy KD at 10 days resembles what is observed normally in animals of 30 days age (Fig S3c).

What was additionally possible to analyze in the frame of revision, however were the levels of presynaptic AZ protein, BRP in 10d control (+*atg5*-RNAi), 30d control and 10d KD (*elav/atg5*-RNAi) animals. As expected (Gupta et al., 2016), aged control flies display an increased immunostaining for BRP, quantitatively very comparable to what we found in KD animals at “already” 10days of age (Fig S8).

2-In Fig 3 the authors assess BRP levels in the entire brain upon knock down of autophagy genes only in MBs. They find BRP is upregulated everywhere. I suppose Refp2 is not upregulated everywhere? There is some data in the supplement on this, but there is no co-labeling of Refp2 with a marker of the cells that express Gal4 (and RNAi to autophagy genes).

- Thanks for raising this point allowing us to clarify this issue. In the first version of the manuscript, we already had shown the p62 aggregation restricted to cell bodies of MB-neurons in *ok107/atg5*-RNAi, *ok107/atg9*-RNAi and *vt30559/atg5*-RNAi (Fig S5). To further confirm and more directly show that

there is a cell autonomous buildup of p62/Ref(2)p aggregates upon KD of autophagy gene, we now performed a co-labeling of p62/Ref(2)p with GFP as a morphological marker of the cells that express Gal4 (and consequently RNAi to the autophagy gene, *atg9*). Indeed, we observe p62/Ref(2)p aggregation specifically in neuron population targeted for *atg*-gene KD (Fig S6). When we used *ok107*-Gal4, a driver expressing in the MB (and a few other neurons), co-labeling showed that the cell bodies positive for GFP clearly corresponded to the p62/Ref(2)p signals (Fig S6). In comparison, *elav*-Gal4, which is a pan-neuronal driver line, in combination with *atg5*-RNAi or *atg9*-RNAi provoked p62 aggregate formation throughout the brain (Fig S6).

3-In Fig 3 the authors block the knock down of Atg5 using Gal80. This is a good control experiment, but an even cooler experiment would be to knock down Atg5 in the entire brain and prevent knock down only in the mushroom bodies. If what the authors propose is correct, this too should rescue the phenotypes (memory and BRP labeling).

- We appreciate that this question was asked. The reviewer certainly poses a valid point here. Unfortunately, creating these genetic combinations followed by isogenization prior to memory and BRP labeling was not possible in the frame of revision.

4-There is less expression of sNPF when autophagy is inhibited. Is there also less RNA expression? How do the authors envisage that lower autophagy causes less sNPF to be expressed? Can this be discussed?

- We thank the reviewer for this valuable and very interesting question. In addition to a significant reduction in amounts of sNPF peptide precursor staining (Fig 5), we now in new experiments determined an about 40%¹ decline of *snpf* transcript levels in morphologically isolated brains of *ok107/atg5*-RNAi using quantitative Real Time-PCR. Thus, as the level of protein and transcript reduction here are obviously comparable, the effect indeed might be transcriptionally mediated, also given that whole-brain quantification of *snpf* transcripts is most likely rather an underestimation of the real effect size (though the MB is the center of *snpf* expression). Notably in mice hypothalamic neurons, an increase or decrease in autophagy induces or suppresses, respectively, the transcription of Neuropeptide Y (NPY; *Drosophila* homolog: sNPF) (Oh et al., 2016). Thus, likely autophagic control over NPY-type neuropeptide transcription is a conserved feature. Based on our new results we now discuss and write, “*We here found that transcript expression level of an NPY family member (sNPF) are controlled by autophagy within the MBs.*”

5-Did the authors try to over express sNPF in Kenyon cells where autophagy genes were knocked down to assess if this rescues metaplasticity?

- This is a very interesting suggestion, we had also considered. However, in preparatory experiments, sheer overexpression of sNPF in MB was actually counter-productive for forming 1h-MTM and particularly 1h-ASM (see below) and thus is not likely to protect autophagy-challenged flies either. It appears likely that uncontrolled constitutive overexpression of sNPF acts negatively, e.g. via receptor desensitization or similar effects.

¹ Relative Fold Change: 0,6417 ± 0.1217, *p<0.05, Paired t-test, n=6

Fig R1

6-I lacked a discussion on how the authors think sNPF results in the regulation of BRP levels?

- We appreciate the opportunity to discuss this intriguing observation in somewhat more detail in manuscript. We write in the discussion: “At this point, we can only speculate concerning the mechanistic basis of this exciting regulation. Interestingly, elevated cAMP signaling is generally driving plasticity in *Drosophila* neurons, while sNPF signaling is meant to reduce cAMP (Vecsey et al., 2014) and thus potentially might be able to reset plastic changes such as increased BRP levels. In apparent contradiction to sNPF signaling directly widely controlling metaplasticity, however, is our finding that MB-specific KD of the sNPF α sufficed to increase BRP levels.”

7-The quality of the western blot in fig 1g is not optimal…

- We thank the reviewer for bringing this to our notice. We have now replaced the blots in the Fig 1g with technically improved versions. In addition to Fig 1g, please find below a comparison of blot images acquired at long and short exposure times respectively. At both the exposure times, an increased accumulation of both p62 and Atg8a can be easily be observed in protein homogenates of *elav/atg5-RNAi* or *elav/atg9-RNAi*, confirming a block in autophagy process. As expected (Hanada et al., 2007), lipidated Atg8a is missing specifically in *atg5* KD but not in *atg9* KD. Due to excessive aggregation of p62 and Atg8a upon pan-neuronal inhibition of autophagy, short exposure time did not capture the signal for p62 and lipidated Atg8a in control samples. Hence, we decided to show the images acquired at high exposure time in the manuscript (Fig 1g).

Fig R2

8-What are the efficiencies of RNA knock down throughout the paper?

- We performed qRT-PCR with the RNA isolated from morphologically isolated fly brains and noticed a significant decline of about 40%² in *atg5* transcript levels in *elav/atg5-RNAi* flies and a significant decline of about 50%³ in *atg9* transcript levels in *elav/atg9-RNAi* compared to age-matched control. Please note that this necessarily is an underestimation of the transcript KD efficacy as *elav-Gal4* targets specifically neurons and not other brain cells, most importantly glia. During RNA isolation and subsequent steps we have transcript contribution from these other cell populations in fly brain, which cause an underestimation of RNA efficiency.

In the manuscript, using MB-specific Gal4-lines, we restricted our manipulations to MB (<2% neurons). In such a situation, we believe that we will not be able to efficiently estimate the RNA KD efficiencies in morphologically isolated fly brains. Therefore, we chose an alternate approach and looked at the accumulation of autophagy markers: p62, Atg8a and Syntaxin-17 in cell bodies of MB neurons in these animals (Fig S4, S5, S6, S7). Suppression of either *atg5* or *atg9* stimulates buildup of p62 and Atg8a in cell bodies of MB neurons but, as expected, did not affect Syntaxin-17 levels, known to operate downstream of autophagosome formation (Itakura and Mizushima, 2013) (Fig S7).

9-Check references to figures because some are not correctly referring to the right panels

- We thank the reviewer for bringing this to our notice. We have now carefully corrected for such errors throughout the manuscript.

Response prepared for Reviewer #2 (Remarks to the Author):

² Relative fold change: 0.5964 ± 0.1034 . ** $p < 0.005$, Paired t-test, $n=7$.

³ Relative fold change: 0.5319 ± 0.1261 . ** $p < 0.005$, Paired t-test, $n=7$.

The mushroom body is a brain center required for the formation and storage of associative memories. In their manuscript, Bhukel et al. report evidence suggesting that a reduction of autophagy in the mushroom body impairs the formation of associative memories. The evidence supporting this conclusion is twofold: first, the authors found that brain-wide and mushroom body specific attenuation of autophagy — which was carried out using the GAL4 system and RNA interference — leads to memory defects; second, the authors observed that attenuation of autophagy in the mushroom body leads to brain-wide increase in the number of presynaptic active zones. They refer to this increase as ‘synaptic metaplasticity’. The authors also suggest that autophagy is necessary to maintain synaptic homeostasis in aging brains and that neuropeptide F — a neuropeptide secreted by the mushroom body — might be a potential non-cell autonomous mediator of this synaptic metaplasticity. The evidence supporting this conclusion includes a comparative study with a hypomorph mutant of neuropeptide F and a knockdown of its receptor. This manuscript proposes an interesting mechanism whereby neuropeptide F could be a key regulator of synaptic homeostasis in the fly brain. However, the experimental evidence supporting the claims required to support such a mechanism is weak and not as thorough as it should and could be. Mainly for this reason, I do not support the publication of this manuscript in Nature Communication. I think the manuscript needs several more experiments and is perhaps better fitted for a more specialized journal. Please find below a more detail criticism of the experiments and the conclusions reached by the authors.

Major points of criticism.

1. Overall effects of the manipulations used to block autophagy To block autophagy, the authors used two transgenic lines that each contains a RNAi construct targeting genes important for autophagy, namely atg5 and atg9. The authors used these tools in combination with the GAL4 expression system to reduce autophagy in the whole brain or in specific subpopulations of neurons. They gaged the effects of these manipulations by measuring the cellular levels of the ubiquitin binding scaffold protein p62, which accumulates when autophagy is blocked. The authors could use more controls than a simple p62 readout to determine how specific the observed defects are. For instance, are the signaling pathways induced during phagocytosis increased as well and are these changes specific to the tissues in which autophagy was manipulated? Are these manipulations causing side effects such as apoptosis? Strong experimental evidence is needed here as this technique — the manipulation of autophagy using these two RNAi constructs — is the technique most experiments are built on and the very foundation of this study.

- We understand the reviewer's concerns regarding the non-autophagy effects of suppressing atg-gene, which also echoed in the critique by reviewer#1. To exclude that our genetic manipulations of autophagy targeting two fundamentally different components of the autophagic machinery is causing considerable side effects of the mentioned kind, we analyzed putative effects on apoptosis on two different ways. Firstly, we immunostained 10d old fly brains for Annexin V and activated Death caspase-1 (Dcp-1) to specifically detect apoptotic cells. Annexin V binds to phosphatidylserine, a marker of apoptosis when it is on the outer leaflet of the plasma membrane (van Genderen et al., 2008). Dcp-1, a *Drosophila* effector caspase, which along with Death-related ICE-like caspase (Drice) is a commonly used marker for cells undergoing apoptosis in *Drosophila* (Sudmeier et al., 2015). As demonstrated previously (Muradian and Schachtschabel, 2001), we noticed

an increased accumulation of apoptotic cells with age in control (Fig S1). Importantly, however, these markers were equally negative after suppression of either *atg*-gene. We now show these data in Fig S1.

Ultrastructurally, apoptosis is typically characterized by a condensation of chromatin close to the inner linings of the nuclear envelope, cytoplasmic shrinkage and active membrane blebbing. We did not notice any of such signs of active apoptosis in the electron micrographs obtained from the autophagy inhibited brains. Moreover, if apoptosis was misregulated in the CNS upon inhibition of autophagy, an altered count of cell bodies should be expected in the 10 day old flies. Thus, we expressed GFP in MB where autophagy had been suppressed using the MB-drivers, *ok107-Gal4* and *vt30559-Gal4*, and manually counted the number of GFP-positive cell bodies in a comparative manner (Fig S2). While average cell body counts were in the expected range, no significant difference was found between control and upon inhibition of autophagy. *+atg9-RNAi*: 1279; *ok107/atg9-RNAi*: 1270; *+atg9-RNAi*: 1327; *vt30559/atg9-RNAi*: 1504 (Fig S2).

- In response to reviewers comment: “*The authors could use more controls than a simple p62 readout to determine how specific the observed defects are*”, we would like to bring to the reviewer’s notice that we used both p62 and Atg8a to confirm a block in autophagy upon pan-neuronal inhibition of either *atg*-gene (Fig 1g). Throughout the paper we restricted our manipulations to specific neurons and confirmed that with immunostaining for p62. We now analyzed additional autophagy markers besides p62: Atg8a and Syntaxin-17 in cell bodies of MB-neurons in *ok107/atg5-RNAi* and *ok107/atg9-RNAi* (Fig S7). Both p62 and Atg8a tend to largely aggregate in cell bodies in *ok107/atg5-RNAi* and *ok107/atg9-RNAi* (Fig S7). Syntaxin-17 acts downstream of *atg5* and *atg9*, KD of either gene should not affect Syntaxin-17 levels (Itakura and Mizushima, 2013). As expected, we did not observe any change in levels of Syntaxin-17 in cell bodies upon MB-specific KD of either *atg*-gene (Fig S7).

2. *Lack of explanation for Table 1 and Table 2. There is no legend for the Table 1 and Table 2 which makes it difficult to interpret the data reported these tables. Table 1 is useless and I cannot interpret the results reported in Table 2. The innate avoidance scores should be normalized and explained.*

- We took the opportunity to provide an extended explanation, which has now been added to the figure legend. Innate odor avoidance scores are represented as an index value that ranges between 0 and 100, with 0 signifying no avoidance and 100 signifying complete avoidance. We would like to emphasize that representing the data as such is generic practice in the field and as we think fair representation of the data.

3. *Subtlety of the learning defects and how they relate to defects seen in aging brains. The effects on short-term and mid-term learning, most of which are reported in Figure 2, are very subtle although statistically significant. Namely, there is a lot of residual learning seen in all manipulated animals. In light of the subtlety of these defects, it would have been worthwhile to explore the learning defects better and more carefully.*

- We appreciate to comment on this point. Principal intention of our analysis was to scrutinize modulations and regulations, which are part of the physiological aging process. Aging in *Drosophila* has been shown previously

to result in specific decline of the ASM component (Tamura et al., 2003; Saitoe et al., 2004; Gupta et al., 2013). Indeed, our genetic impairment of autophagy in the MB provoked a qualitatively and quantitatively similar decrease of ASM (Fig 2) as observed in aged brains. Both MTM and STM comprise of an ARM and an ASM component (Knapek et al., 2011; Bouzaiane et al., 2015). So when ASM (the age-sensitive component) declines, ARM stays intact, explaining the residual learning MTM scores. Thus, we are convinced that our findings are relevant in the context of age-induced, physiological decline of memory formation, at least in the fruit fly.

b. Unfortunately, mass training of population of flies reported as a preference index is a very limited analysis. Because the learning deficits caused by the attenuation of autophagy in the mushroom body is the major finding made by this study, the experimental evidence supporting this claim needs to be stronger.

- We are afraid we have to politely disagree in this point. Aversive olfactory conditioning is a widely accepted Pavlovian conditioning protocol in the field of *Drosophila* learning and memory. Right from the first study (Quinn et al., 1974), which described the paradigm to measure olfactory (and visual) memory, to the studies demonstrating age induced memory impairment (Tully and Quinn, 1985; Saitoe et al., 2005; Horiuchi and Saitoe, 2005), mass training of *Drosophila* populations has been used. Thus, the whole scientific context of our study has been exclusively using the very same paradigm as used for other work. We, however, appreciate the reviewers concern that per se such “group” data might miss aspects of learning on the individual level. Notably, in response to similar criticisms, Tully (1986) conducted a series of experiments that compared mass training with individual training procedures. This study demonstrated that 1. an individual’s behavior is not fundamentally influenced by other flies, and 2. the probability of making a correct choice is similar among individuals that have been group trained (Tully, 1986). Based on these findings, Tully and his colleagues have cogently argued that the assessment of learning in individuals is not essential for the appropriate investigation of genetic issues (Tully, 1986; Tully and Gergen, 1986). Thus, we see mass training as performed in our analysis as the appropriate strategy for our questions in the moment. This does not exclude that future studies might involve training procedures on individual fly level. As now, such procedures to the best of our knowledge have never been used in the context of age-induced memory impairment, and are extensive and established at only very few places.

c. Also, the authors mentioned that “KD of a core autophagy component within the MBs suffices to mimic the usual age-induced specific decay of the ASM component” but they did not compare the learning defects they observed to the ones normally seen in aged flies. This criticism applies to all the other experiments as well.

- We thank the reviewer for having put our attention on this point. We previously showed that aged flies display reduced MTM scores tested 3 hours after training (Gupta et al., 2013). We now performed an additional experiment where we directly compared MTM scores of 10d control (+/*atg5*-RNAi), 30d control and 10d KD animals (*elav/atg5*-RNAi) tested 1 hour after training. Indeed, the 1h-MTM scores of 10d KD animals were very comparable to the values of 30d control animals (Fig S3). Notably, *elav/atg5*-

RNAi animals did not survive in sufficient numbers to allow for testing 1h MTM at 30 days of age.

Testing ARM and ASM for 30 day old flies requires rearing large number of flies in repetitive rounds and unfortunately was impossible in the frame of such a revision period. However, in our previous studies, we and others repeatedly showed that it is the ASM component which is sensitive to aging (means at 30 days) (Gupta et al., 2013). As KD of autophagy with *elav-Gal4* specifically affects the ASM at 10 days (Fig 2, i-l)., we can be sure that autophagy KD at 10 days resembles what is observed normally in animals of 30 days age. In the frame of revision, however we could now directly compare the levels of presynaptic AZ protein, BRP in 10d control (+/*atg5*-RNAi), 30d control and 10d KD (*elav/atg5*-RNAi) animals. As expected (Gupta et al., 2016), aged control flies display an increased immunostaining for BRP, quantitatively very comparable to what we found in KD animals at “already” 10days of age (Fig S8).

4. Changes in the number of presynaptic zones. The authors claim observing an increase in the number presynaptic zones — as measured by immunostaining using an antibody against bruchpilot, a master regulator of presynaptic zones — across the brain when autophagy is abolished only in the mushroom body. They refer to this phenomenon as ‘presynaptic metaplasticity’. The evidence supporting this conclusion is rather weak. The comparison of the overall fluorescence of whole brains are provided as the major piece of evidence. Immunostaining on whole brains can be extremely variable and a more detailed study would be helpful to make this evidence stronger. Other evidence includes electron micrographs, which also show very subtle defects. The authors need stronger and more detailed evidence — including functional evidence — to support the claim that abolishing autophagy in the mushroom body affects the number of presynaptic zones across the entire brain.

- We appreciate the opportunity to summarize our findings concerning the brain-wide presynaptic metaplasticity. We for this question used three technically independent approaches: (1) Quantification of BRP immunofluorescence signals throughout the central brain (Fig 3, S9), (2) Electron microscopic analysis obtained from the genetically targeted (MB lobe synapses) and non-targeted region (projection neuron bouton synapses in MB calyx) (Fig 4) and (3) super-resolution STED microscopy in non-targeted region (Calyx and Antennal Lobes; Fig 4). The scenario which in this manuscript we describe after genetically compromising autophagy in both qualitative as well as quantitative terms resembles the synaptic scenario we before described for aged flies. Notably, in our previous work, we used genetic manipulations to mimic this metaplasticity (4 gene copies of BRP) and showed that this is by itself sufficient to cause ASM deficits in *Drosophila* (Gupta et. al. 2016). This finding has been reproduced by the Gerber lab recently (Michels et al., 2018). Moreover, in our previous work, we had shown that age-induced presynaptic metaplasticity is associated with increased synaptic vesicle release using an optophysiological assay (Gupta et. al. 2016). Thus, concerning the current status of analysis, we have provided evidence that the synaptic status after having genetically attenuated autophagy indeed resembles the synaptic situation of the aged brain.
- In response to the reviewer's question, we now included new data concerning AZ densities normalized to synaptic bouton area retrieved from electron micrographs (Fig S11). We indeed find a slightly increased area normalized density of T-bars upon MB-specific attenuation of autophagy. We, however,

cannot fully exclude that the increase in average T bar size might have facilitated T bar detection as such and have contributed to this effect. .

5. *Fast synaptic transmission is not responsible for the observed synaptic metaplasticity* The data supporting the claims made in lines 259 to 272 — that synaptic metaplasticity is not a consequence of increased or decreased neuronal activity in the mushroom body — should be provided as a supplemental figure.

- We have now provided the requested data as supplementary figure S12.

5. *Neuropeptide F as a regulator of ‘presynaptic metaplasticity’*

a. *Lastly, the authors report evidence showing that blocking autophagy reduces the production of neuropeptide F made by the mushroom body.*

- We found that blocking autophagy in MB reduces the *snpf* transcript levels by about 40%⁴ (0.6097 ± 0.1438) in samples from morphologically isolated brain. This now offers a direct explanation for the 50% reduction in immunostaining for sNPF peptide precursor levels in the MB we observed in immunostainings (Fig 5).

b. *The authors also found that an hypomorph mutant of neuropeptide F and a mushroom-body specific knockdown of the neuropeptide F receptor both show an increase in synaptic metaplasticity, similarly as observed in the brain of flies in which autophagy was blocked in the mushroom body only. Why the knockdown of the receptor in the mushroom body only is sufficient to see these effects — and not in all neurons — is not discussed.*

- The reviewer here certainly mentions a very interesting question. We on purpose had included these data as they per se demonstrate the importance of sNPF signaling for the generic synaptic status of the *Drosophila* brain. Indeed, it is surprising that the KD of sNPF-receptor in the MB only suffices to provoke synaptic changes clearly extending over the MB. We show above that unregulated sNPF release is rather counterproductive for preserving metaplasticity associated ASM (Fig. R1). Potentially, sNPF-receptor signaling within the MB might be important to control sNPF secretion in a physiological manner via a quasi autocrine mechanism. We now write: “In the moment, we can only speculate as to why KD of sNPF-receptor also results in extended metaplastic changes. Potentially, sNPF-receptor signaling within the MB might be important to control sNPF secretion in a physiological manner via a quasi autocrine mechanism.”

c. *The authors suggest, but do not show, that neuropeptide F is important to maintain presynaptic metaplasticity in aging brain, namely by maintaining homeostasis in the number of presynaptic zones. They suggest that a reduction in neuropeptide F might occur in older animals and be the cause age related learning defects. This last claim is very confusing because not properly explained as well as not properly tested, not directly tested.*

⁴ Relative Fold Change: $0,6417 \pm 0.1217$, * $p < 0.05$, Paired t-test, n=6

- We are thankful for this suggestion. In earlier version of manuscript we suggested (but did not test) that aged flies should suffer from a decline in sNPF. We now tested this hypothesis. Indeed, we observed a significant decline in sNPF peptide precursor levels with age in *Drosophila* central brain (Fig 5, h-q).

Bibliography

- Bjorkoy, G., T. Lamark, S. Pankiv, A. Overvatn, A. Brech, and T. Johansen. 2009. Monitoring autophagic degradation of p62/SQSTM1. *Methods Enzymol.* 452:181–197. doi:10.1016/S0076-6879(08)03612-4.
- Bouzaiane, E., S. Trannoy, L. Scheunemann, P.-Y. Placais, and T. Preat. 2015. Two independent mushroom body output circuits retrieve the six discrete components of *Drosophila* aversive memory. *Cell Rep.* 11:1280–1292. doi:10.1016/j.celrep.2015.04.044.
- van Genderen, H.O., H. Kenis, L. Hofstra, J. Narula, and C.P.M. Reutelingsperger. 2008. Extracellular annexin A5: functions of phosphatidylserine-binding and two-dimensional crystallization. *Biochim. Biophys. Acta.* 1783:953–963. doi:10.1016/j.bbamcr.2008.01.030.
- Gupta, V.K., U. Pech, A. Bhukel, A. Fulterer, A. Ender, S.F. Mauermann, T.F.M. Andlauer, E. Antwi-Adjei, C. Beuschel, K. Thriene, M. Maglione, C. Quentin, R. Bushow, M. Schwärzel, T. Mielke, F. Madeo, J. Dengjel, A. Fiala, and S.J. Sigrist. 2016. Spermidine Suppresses Age-Associated Memory Impairment by Preventing Adverse Increase of Presynaptic Active Zone Size and Release. *PLoS Biol.* 14:e1002563. doi:10.1371/journal.pbio.1002563.
- Gupta, V.K., L. Scheunemann, T. Eisenberg, S. Mertel, A. Bhukel, T.S. Koemans, J.M. Kramer, K.S.Y. Liu, S. Schroeder, H.G. Stunnenberg, F. Sinner, C. Magnes, T.R. Pieber, S. Dipt, A. Fiala, A. Schenck, M. Schwaerzel, F. Madeo, and S.J. Sigrist. 2013. Restoring polyamines protects from age-induced memory impairment in an autophagy-dependent manner. *Nat. Neurosci.* 16:1453–60. doi:10.1038/nn.3512.
- Hanada, T., N.N. Noda, Y. Satomi, Y. Ichimura, Y. Fujioka, T. Takao, F. Inagaki, and Y. Ohsumi. 2007. The Atg12-Atg5 conjugate has a novel E3-like activity for protein lipidation in autophagy. *J. Biol. Chem.* 282:37298–37302. doi:10.1074/jbc.C700195200.
- Horiuchi, J., and M. Saitoe. 2005. Can flies shed light on our own age-related memory impairment? *Ageing Res. Rev.* 4:83–101. doi:10.1016/j.arr.2004.10.001.
- Itakura, E., and N. Mizushima. 2013. Syntaxin 17: the autophagosomal SNARE. *Autophagy.* 9:917–919. doi:10.4161/auto.24109.
- Knapek, S., S. Sigrist, and H. Tanimoto. 2011. Bruchpilot, a synaptic active zone protein for anesthesia-resistant memory. *J. Neurosci.* 31:3453–3458. doi:10.1523/JNEUROSCI.2585-10.2011.
- Michels, B., H. Zwaka, R. Bartels, O. Lushchak, K. Franke, T. Endres, M. Fendt, I. Song, M. Bakr, T. Budragchaa, B. Westermann, D. Mishra, C. Eschbach, S. Schreyer, A. Lingnau, C. Vahl, M. Hilker, R. Menzel, T. Kahne, V. Lessmann, A. Dityatev, L. Wessjohann, and B. Gerber. 2018. Memory enhancement by ferulic acid ester across species. *Sci. Adv.* 4:eaat6994. doi:10.1126/sciadv.aat6994.
- Muradian, K., and D.O. Schachtschabel. 2001. The role of apoptosis in aging and age-related disease: update. *Z. Gerontol. Geriatr.* 34:441–446.
- Oh, T.S., H. Cho, J.H. Cho, S.-W. Yu, and E.-K. Kim. 2016. Hypothalamic AMPK-induced autophagy increases food intake by regulating NPY and POMC expression. *Autophagy.* 12:2009–2025. doi:10.1080/15548627.2016.1215382.
- Pankiv, S., T.H. Clausen, T. Lamark, A. Brech, J.-A. Bruun, H. Outzen, A. Overvatn,

- G. Bjorkoy, and T. Johansen. 2007. p62/SQSTM1 binds directly to Atg8/LC3 to facilitate degradation of ubiquitinated protein aggregates by autophagy. *J. Biol. Chem.* 282:24131–24145. doi:10.1074/jbc.M702824200.
- Quinn, W.G., W.A. Harris, and S. Benzer. 1974. Conditioned behavior in *Drosophila melanogaster*. *Proc. Natl. Acad. Sci. U. S. A.* 71:708–712.
- Saitoe, M., J. Horiuchi, T. Tamura, and N. Ito. 2005. *Drosophila* as a novel animal model for studying the genetics of age-related memory impairment. *Rev. Neurosci.* 16:137–149.
- Saitoe, M., T. Tamura, J. Horiuchi, and N. Ito. 2004. [Identification of the memory component that decays with age in *Drosophila*]. *Nihon Shinkei Seishin Yakurigaku Zasshi.* 24:231–237.
- Sudmeier, L.J., S.P. Howard, and B. Ganetzky. 2015. A *Drosophila* model to investigate the neurotoxic side effects of radiation exposure. *Dis. Model. Mech.* 8:669–677. doi:10.1242/dmm.019786.
- Tamura, T., A.-S. Chiang, N. Ito, H.-P. Liu, J. Horiuchi, T. Tully, and M. Saitoe. 2003. Aging specifically impairs amnesiac-dependent memory in *Drosophila*. *Neuron.* 40:1003–1011.
- Tully, T. 1986. Measuring learning in individual flies is not necessary to study the effects of single-gene mutations in *Drosophila*: a reply to Holliday and Hirsch. *Behav. Genet.* 16:449–455.
- Tully, T., and J.P. Gergen. 1986. Deletion mapping of the *Drosophila* memory mutant amnesiac. *J. Neurogenet.* 3:33–47.
- Tully, T., and W.G. Quinn. 1985. Classical conditioning and retention in normal and mutant *Drosophila melanogaster*. *J. Comp. Physiol. A.* 157:263–277.

REVIEWERS' COMMENTS:

Reviewer #1 (Remarks to the Author):

All my points have been adequately addressed.

Reviewer #2 (Remarks to the Author):

I have read the detailed and very thorough response to our critiques. I am satisfied with the new data provided as well as the clarification given. I believe this manuscript has significantly improved overall and is suitable for publication. I am happy to support its publication in Nature Communications.

ADDITIONAL EDITORIAL REQUESTS:

Your manuscript has been checked for clarity and against journal policies and formatting style. The issues listed below must be addressed. If using Microsoft Word, please use the tracked changes feature to make these changes.